# You Can Trust Your Clustering Model: A Parameter-free Self-Boosting Plug-in for Deep Clustering

**Hanyang Li**[1], **Yuheng Jia**[1,2,3]*, **Hui Liu**[3], **Junhui Hou**[4]*

[1]School of Computer Science and Engineering, Southeast University, Nanjing 210096, China
[2]Key Laboratory of New Generation Artificial Intelligence Technology and Its
Interdisciplinary Applications (Southeast University), Ministry of Education, China
[3]School of Computing Information Sciences, Saint Francis University, Hong Kong, China
[4]Department of Computer Science, City University of Hong Kong, Hong Kong, China
{lihanyang, yhjia}@seu.edu.cn, h2liu@sfu.edu.hk, jh.hou@cityu.edu.hk

## Abstract

Recent deep clustering models have produced impressive clustering performance. However, a common issue with existing methods is the disparity between global and local feature structures. While local structures typically show strong consistency and compactness within class samples, global features often present intertwined boundaries and poorly separated clusters. Motivated by this observation, we propose **DCBoost**, **a parameter-free plug-in** designed to enhance the global feature structures of current deep clustering models. By harnessing reliable local structural cues, our method aims to elevate clustering performance effectively. Specifically, we first identify high-confidence samples through adaptive $k$-nearest neighbors-based consistency filtering, aiming to select a sufficient number of samples with high label reliability to serve as trustworthy anchors for self-supervision. Subsequently, these samples are utilized to compute a discriminative loss, which promotes both intra-class compactness and inter-class separability, to guide network optimization. Extensive experiments across various benchmark datasets showcase that our DCBoost significantly improves the clustering performance of diverse existing deep clustering models. Notably, our method improves the performance of current state-of-the-art baselines (e.g., ProPos) by more than 3% on average and amplifies the silhouette coefficient by over $7\times$. Code is available at https://github.com/l-h-y168/DCBoost.

## 1 Introduction

Deep clustering aims to use deep neural networks to uncover the intrinsic structure of data by partitioning samples into groups based on their similarity, without relying on any class labels. Early methods employed autoencoders [12, 37] to extract data representations, which significantly enhanced clustering performance and helped establish deep clustering as a prominent research field. More recently, the integration of self-supervised learning techniques [11, 4, 9] has further advanced deep clustering, leading to impressive clustering performance [36, 20, 13].

Despite the impressive performance, we observe that existing methods do not construct a reliable global structure, but all build a reliable local pattern. For example, as shown in Fig. 1(a), the intra-class similarity is relatively low, and the inter-class similarity remains noticeable, suggesting

---

*Corresponding author.

39th Conference on Neural Information Processing Systems (NeurIPS 2025).

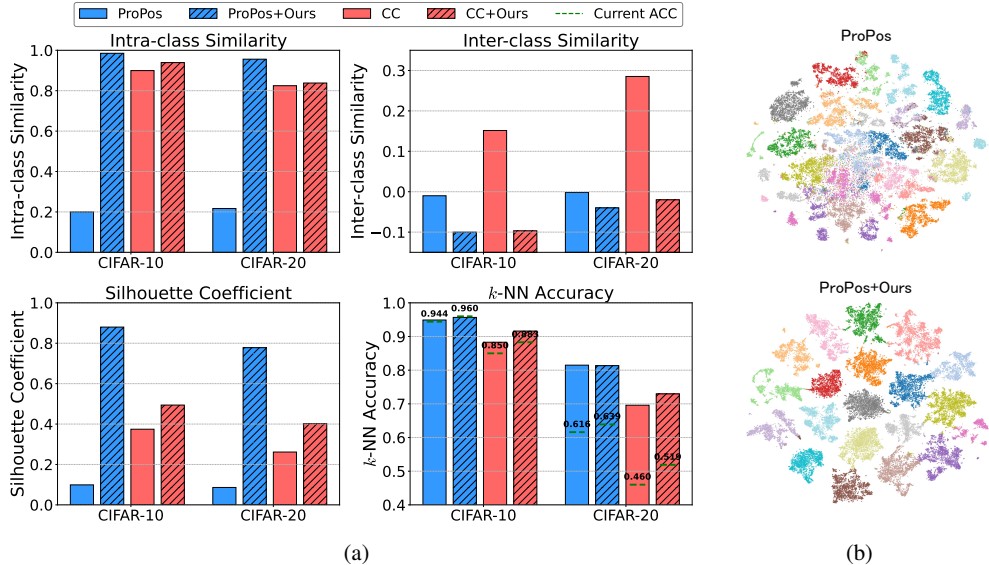

Figure 1: (**a**) Quantitative comparisons of the global structure in terms of intra-class similarity, inter-class similarity, and silhouette coefficient, and the local structure in terms of $k$-NN accuracy on CIFAR-10 and CIFAR-20. The original model ProPos and CC show weak global structure but strong local structure. Our method maintains their high local accuracy while significantly improving the global structure and overall clustering performance. (**b**) T-SNE visualization of clustering without (upper) and with (bottom) applying our method on CIFAR-20. Initially, the existing model exhibits severe overlap between different classes, leading to poor separability. After boosting, class boundaries become significantly clearer and more distinguishable.

that samples within clusters lack compactness, while the separation between clusters is also inadequate. This observation is further supported by their low silhouette coefficient (SC), reflecting poor class separability, unclear decision boundaries, and entangled feature spaces—indicating a general weakness in capturing global structural information. Furthermore, when examining local structural quality with the $k$-NN accuracy (i.e., the proportion of a sample's $k$-nearest neighbors ($k$-NNs) that share the same label), this metric remains consistently high across different deep clustering models, often matching or exceeding clustering accuracy, which means that the feature space still preserves reliable local pattern where semantically similar samples are closely grouped. Additionally, the t-SNE visualization in Fig. 1(b) shows that while some rough cluster formations are observable, the overall distribution remains entangled, with many clusters overlapping or lacking clear margins. However, within each loosely formed region samples with similar features tend to group together locally, even if the overall class boundaries are ambiguous. This observation further reinforces our assertion that while existing models face challenges in capturing global semantics, they tend to preserve dependable local patterns.

Building on the above observations, we propose a **parameter-free plug-in** that leverages trustworthy local structural cues to guide the learning of global feature structure and accordingly improve the clustering performance. Specifically, we first propose a dynamic strategy to identify high-confidence samples that reflect reliable local structures. These samples are then used to construct a discriminative loss, which enhances intra-class compactness, promotes inter-class separability, and preserves instance-level consistency. As shown in Fig. 1(a), after applying our method, the global structure and clustering performance improve significantly. For ProPos on CIFAR-10, the silhouette coefficient increases from **0.10** to **0.74**, while the clustering accuracy (ACC) improves from **94.4%** to **96.0%**. Extensive experiments on five benchmark datasets demonstrate that our method substantially improves the clustering performance of six existing deep clustering models, i.e., our method improves the average performance of current state-of-the-art baseline ProPos by more than 3%.

In summary, our contributions are as follows.

- We are the first to observe that many deep clustering methods exhibit poor global structure while retaining trustworthy local structures, revealing an important phenomenon previously overlooked.

Based on this sight, we leverage these reliable local cues to guide the learning of global feature structure and accordingly improve overall clustering performance.

- We propose DCBoost, a **parameter-free plug-in** integrating an adaptive $k$-NN filtering to select high-confidence samples and a discriminative loss that encourages intra-class compactness and inter-class separation, all without requiring manually tuned hyperparameters.

- Extensive experiments on five benchmark datasets and six deep clustering models demonstrate the effectiveness, universality, and zero-cost nature of DCBoost.

## 2 Related Work

**Deep clustering** aims to learn data representations through deep neural networks while leveraging these learned features to guide clustering. Early methods [12, 37, 10, 29] utilized autoencoders for feature learning, followed by joint training of features and clustering, forming the earliest paradigm of deep clustering. With the rise of self-supervised learning, contrastive [11, 4, 18, 2] and non-contrastive [9, 5] paradigms have been introduced into deep clustering, giving rise to two main branches: **representation-based clustering** [35, 32, 13, 39, 23] and **clustering-head-based clustering** [36, 20, 7, 27, 25]. Representation-based methods typically apply classical clustering algorithms (e.g., $k$-means) on features to generate pseudo labels, which are then fed back as supervision to improve representation learning. In contrast, clustering-head-based methods attach a dedicated classification head to the network, allowing pseudo labels to be directly predicted and optimized in an end-to-end fashion. Most existing deep clustering models employ strategies such as contrastive learning [20, 13], mutual information maximization [38], neighborhood consistency [7, 40] to enhance performance. Additionally, heuristic filtering techniques are introduced to remove noisy pseudo labels [36, 27, 30, 16], relying on classification confidence or consistency between predictions. Both paradigms leverage self-supervised learning to produce clustering-friendly representations and have significantly advanced the performance. Recently, large-scale vision and multimodal models like CLIP have provided new insights for deep clustering such as [1, 22], incorporating external knowledge and improving clustering performance.

**Motivation.** Despite notable progress, we empirically observe that the global structure of existing models may be unreliable with noisy semantics and weak class separability, while local neighborhoods tend to be more stable and reliable. Motivated by this, we propose a dynamic sample selection method extracting high-confidence samples via local consistency to guide global structure optimization. Our method is parameter-free and applies to both representation- and clustering-head-based models.

## 3 Proposed Method

**Overview.** Given an unlabeled dataset $\mathbb{X} = \{x_i\}_{i=1}^n$ containing $n$ unlabeled samples belonging to $c$ semantic clusters, deep clustering aims to group these samples into $c$ different clusters. The proposed DCBoost is a generic plug-and-play algorithm for boosting existing deep clustering models. As illustrated in Fig. 2, we initialize DCBoost by adopting any pre-trained deep clustering model as the target network $f_t(\cdot)$ and duplicating it to construct the online network $f_o(\cdot)$, followed by a randomly initialized non-

---

**Algorithm 1** The proposed algorithm DCBoost

**Require:** Input data $X$, pre-trained existing deep clustering model $\mathcal{M}$
1: Initialize $f_o(\cdot)$ and $f_t(\cdot)$ with $\mathcal{M}$, and initialize $g(\cdot)$ randomly
2: **while** Clustering **do**
3:     Apply $k$-means on the output of $f_t(x)$ to assign pseudo-labels $y$
4:     **for** $b = 1$ to $N/B$ **do**
5:         Randomly augment $x$
6:         Select high-confidence samples using adaptive $k$-NN by Eq. (2)
7:         Compute the discriminative loss using Eqs. (3), (4), (6), and (8)
8:         Update $g(f_o(\cdot))$ and $f_t(\cdot)$ with the SGD optimizer and exponential moving average, respectively
9:     **end for**
10: **end while**
11: Apply $k$-means on the output of $f_t(x)$ for final clustering

---

linear predictor $g(\cdot)$. We apply two types of weak augmentations $\mathcal{T}^1(\cdot)$ and $\mathcal{T}^2(\cdot)$ to each sample to generate two different views $\mathcal{T}^1(x_i)$ and $\mathcal{T}^2(x_i)$, which are encoded into $L2$- normalized feature vectors $z_o$ and $z_t$ by $f_o(\cdot)$ and $f_t(\cdot)$. For each training batch $B$ of size $n_B$, DCBoost selects a set of high-confidence samples using an adaptive $k$-NN method (Sec. 3.1, local structure mining), which are further utilized to construct a discrimination loss to fine-tune the model (Sec. 3.2, local-guided global refinement). To obtain pseudo-labels for all samples $\mathbb{Y} = \{y_i\}_{i=1}^n$, we apply $k$-means clustering on the output of the target network at the end of each training epoch for all samples (i.e., the inference branch of Fig. 2). Algorithm 1 lists the overall process of our DCBoost.

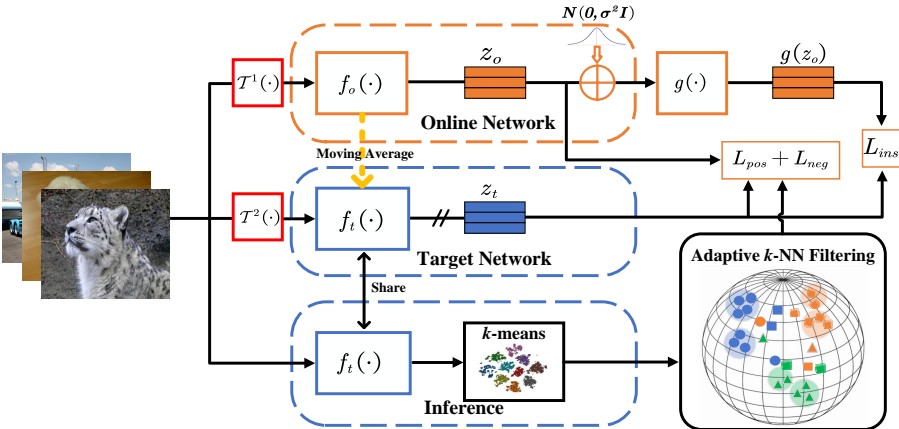

Figure 2: Illustration of our DCBoost framework. During training, the gradient backpropagation of the target network is detached, and the parameters of $f_t(\cdot)$ are updated using exponential moving average (EMA) from those of $f_o(\cdot)$. The overall discriminative loss contains three terms: positive loss $L_{pos}$, negative loss $L_{neg}$, and instance consistency loss $L_{ins}$. The inference branch outputs the pseudo labels of all samples at the end of each epoch.

## 3.1 High-Confidence Sample Selection Leveraging Local Structure via Adaptive $k$-NN Filtering

As previously discussed, empirical evidence indicates that existing deep clustering models excel at capturing local structure, which can be used to refine the global feature space to improve the overall clustering performance. However, it is crucial to acknowledge that not all local structures are inherently dependable. Local regions, particularly near cluster boundaries, can harbor noisy or ambiguous data points. Thus, it becomes imperative to discern and utilize only the most reliable samples–those exhibiting clear structural coherence–to provide stable guidance for training.

To this end, we propose a parameter-free approach to extract high-confidence samples by leveraging neighborhood label agreement. Specifically, for each sample's feature $z_i$ (i.e., the features obtained through the online and target networks, and Appendix. A provides more details), we retrieve its $k$ nearest neighbors and validate their pseudo-labels. A sample is deemed high-confidence only if all $k$ neighbors share the same pseudo-label with $z_i$: $y_i = y_j, \forall j \in \mathbb{N}_k(z_i)$, where $\mathbb{N}_k(z_i)$ is the neighbor set for $z_i$. Conversely, if any neighbor exhibits a different pseudo-label, the sample likely resides near a class boundary, and should be excluded from high-confidence sample set. However, determining the value of $k$ is a challenging task, as the quality of local structures may vary significantly across datasets, models, and training stages, influenced by the number of underlying classes. A small $k$ will include more samples in the high-confidence set but may reduce the average correctness of the pseudo labels due to looser agreement criteria. Conversely, a large $k$ ensures higher label consistency and thus reliability, but may result in fewer selected samples, potentially weakening the overall self-supervision signal. Figs. 3(a) and (b) illustrate this phenomenon on three typical datasets.

To solve the above issue, we propose an adaptive determination method to better control the number and quality of high-confidence samples across different training batches. Specifically, we first define a set of candidate values for $k$, denoted as $\mathbb{K} = \{k_s \mid k_s = s, \ s = 1, 2, \ldots, m\}$. In all experiments, we set $m = 50$ to ensure a comprehensive search space while maintaining computational efficiency. Then, for each $k_s$, we compute the corresponding $\text{score}_{k_s}$ as:

$$\text{score}_{k_s} = k_s \times \frac{n_s}{n_B}, \tag{1}$$

where $n_s = \sum_{i=1}^{n_B} \mathbb{I}(y_i = y_j, \forall z_j \in \mathbb{N}_{k_s}(z_i))$ is the number of selected high-confidence samples, given $k_s$. $\mathbb{I}$ is an indicator function that returns 1 if the condition is met and 0 otherwise. We finally select the candidate yielding the highest score as the value of $k$.

**Remark.** Geometrically, the score in Eq. 1 can also be interpreted as the area of the shaded rectangle as illustrated in Fig. 3(b). Specifically, the $x$-axis represents the number of neighbors $k$, and the

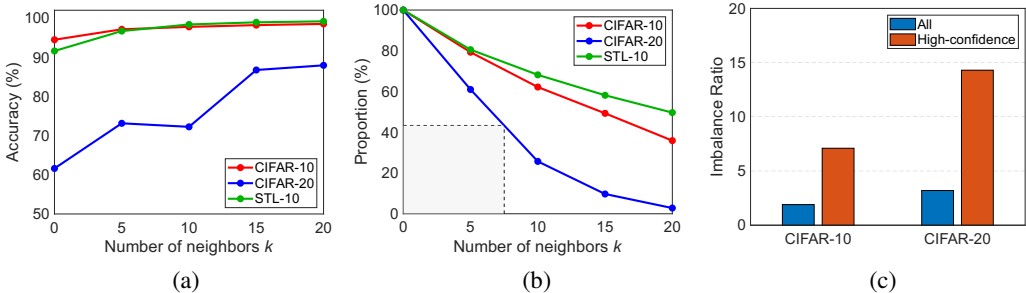

(a)             (b)             (c)

Figure 3: Impact of the value of $k$ on (**a**) the true accuracy of high-confidence samples, and (**b**) the proportion of high-confidence samples, i.e., the ratio of selected high-confidence samples to the total number of samples in a batch under given $k$. A smaller $k$ may reduce the reliability of the selection, whereas a larger $k$ generally improves sample quality but selects fewer samples. (**c**) Comparison of imbalance ratios between high-confidence samples and all samples across classes. The number of high-confidence samples selected per class can be imbalanced, without proper constraints, classes with larger sample counts may dominate training and negatively affect performance.

$y$-axis represents the corresponding proportion $\frac{n_s}{n_B}$. The score thus corresponds to the rectangular area surrounded by them. This intuitive perspective helps to understand the trade-off being optimized.

After determining the value of $k$, the high-confidence samples set $X_h$ will be computed by:

$$\mathbb{X}_h = \{x_i \in B | y_i = y_j, \forall z_j \in \mathbb{N}_{k_s}(z_i)\}. \tag{2}$$

Our method strikes a well trade-off between the quality and quantity of self-supervision signals, which allows the model to efficiently exploit abundant self-supervision early, while later focusing on highly confident samples, ultimately improving clustering performance. This trade-off is further demonstrated by the variation of selected $k$ values during training shown in Appendix C, Fig. 9.

## 3.2 Global Structure Refinement via Pseudo Label-augmented Discriminative Loss

We propose to refine the global feature representation by fine-tuning the model with the following discriminative loss:

$$L = L_{pos} + L_{neg} + L_{ins}, \tag{3}$$

where $L_{pos}$, $L_{neg}$, and $L_{ins}$ are positive loss term, negative loss term, and instance consistency loss term, respectively. In each batch, $L_{pos}$ and $L_{neg}$ are applied to high-confidence samples to enhance the clustering quality, while $L_{ins}$ is applied to all samples to maintain instance-level consistency.

**Positive loss $L_{pos}$.** To enhance intra-class compactness, $L_{pos}$ brings together high-confidence samples with identical pseudo-labels through the formula:

$$L_{pos} = \frac{1}{2c_B} \sum_{c=1}^{c_B} \sum_{i,j \in \mathbb{X}_c} w_c d_{ij}^2. \tag{4}$$

Here, $c_B$ represents the number of distinct classes among high-confidence samples set within a batch $B$, and $\mathbb{X}_c \subset \mathbb{X}_h$ represents the set of samples with pseudo-label $c$ in the high-confidence set $\mathbb{X}_h$. The term $d_{ij}^2 = \|z_i^o - z_j^t\|_2^2 = 2 - 2z_i^o z_j^t$ signifies the squared Euclidean distance between the outputs of the online and target networks for samples $i$ and $j$ sharing the same pseudo-label. Furthermore, as depicted in Fig. 3(c), we notice a concentration of high-confidence samples in specific classes, i.e., the dominant class will have much more number of samples than the nondominant ones, leading to the pseudo class imbalance problem, which can potentially bias the fine-tuning process. Hence, we introduce a normalization coefficient $w_c$ to reduce the impact of class imbalance during optimization:

$$w_c = \frac{1}{\|\sum_{x \in \mathbb{X}_c} z^o\|_2 \|\sum_{x \in \mathbb{X}_c} z^t\|_2}. \tag{5}$$

Specifically, $w_c$ can be seen as the inverse of the square of the $L2$ norms of the aggregated features from class $c$. When a class has more high-confidence samples (thus a larger summed feature norm), its influence on the loss is automatically reduced. In this way, the loss naturally compensates for the imbalance in the number of high-confidence samples across classes. As a summary, the positive loss in Eq. 4 enforces intra-class compactness on high-confidence samples and introduces the weight

coefficient to balance class-wise contributions, refining the feature space and improving overall clustering performance.

**Negative loss $L_{neg}$.** To increase inter-class separability, we repel $c_B$ cluster prototypes derived from high-confidence samples in each batch. The proposed inter-class separation loss is formulated as:

$$L_{neg} = -\sum_{c_1=1}^{c_B} \sum_{c_2=1}^{c_B} \|v_{c_1}^o - v_{c_2}^t\|_2^2 (c_1 \neq c_2) = \sum_{c_1=1}^{c_B} \sum_{c_2=1}^{c_B} (-2 + 2v_{c_1}^o v_{c_2}^t (c_1 \neq c_2)), \qquad (6)$$

where $v^o$ and $v^t$ represent the $L2$-normalized prototypes computed from the online networks $z^o$ and target networks $z^t$, respectively:

$$v_{c_1}^o = \frac{\sum_{x \in \mathbb{X}_{c_1}} z^o}{\|\sum_{x \in \mathbb{X}_{c_1}} z^o\|_2}, v_{c_2}^t = \frac{\sum_{x \in \mathbb{X}_{c_2}} z^t}{\|\sum_{x \in \mathbb{X}_{c_2}} z^t\|_2}. \qquad (7)$$

By minimizing the loss in Eq. 6, our method encourages the inter-class separation through Euclidean distance-based repulsion between prototypes.

**Instance consistency loss $L_{ins}$.** To preserve consistency of an instance in the online network and target network, $L_{ins}$ aligns augmented views of the same instance:

$$L_{ins} = \left\|g\left(f^o\left(\mathcal{T}^1(x)\right) + \sigma\varepsilon\right) - f^t\left(\mathcal{T}^2(x)\right)\right\|_2^2, \text{where } \varepsilon \sim N(0, I), \qquad (8)$$

where $\varepsilon$ represents Gaussian noise sampled from a normal distribution $N(0, I)$ define $I$ here the identity matrix, and $\sigma$ controls its intensity, which is set to 0.001 in all experiments. We treat the feature-space vicinity of one augmented view as positive samples for the other view, assuming they share the same semantics. By minimizing the loss in Eq. 8, out method stabilizes global structure and prevents representation collapse.

## 4 Experiment

### 4.1 Experiment Settings

**Datasets, backbones and baselines.** We evaluated our method on five widely used benchmark datasets, including CIFAR-10, CIFAR-20 [17], STL-10 [6], ImageNet-10, and ImageNet-Dogs [3], as summarized in Table 1. To ensure a fair evaluation of our boosting effect, we used the same image size and backbone architectures as in the

Table 1: Summary of datasets.

| Dataset | Split | #Samples | #Classes |
|---|---|---|---|
| CIFAR-10 | Train+Test | 60,000 | 10 |
| CIFAR-20 | Train+Test | 60,000 | 20 |
| STL-10 | Train+Test | 13,000 | 10 |
| ImageNet-10 | Train | 13,000 | 10 |
| ImageNet-Dogs | Train | 19,500 | 15 |

original settings of the respective baseline methods. We integrated DCBoost into six existing models, including three representation-based methods, i.e., BYOL [9], CoNR [39], and ProPos [13], and three clustering-head-based methods: CC [20], SCAN [36], and CDC [16]. In addition to improving existing models, we compared our method with other deep clustering models, including NNM [7], GCC [40], IDFD [34], TCL [21], TCC [32], SPICE [27], SeCu [31], and DPAC [38] to provide a comprehensive evaluation. More implementation details are shown in Appendix. A.

### 4.2 Main Results

**Significant and consistent improvement on different deep clustering models.** As shown in Table 2, our method consistently improves existing models' performance. Notably, models with lower baseline performance, such as BYOL and CC, benefit significantly from our method, with their performance improving by 3.1% and 4.6%, respectively, greatly enhancing their competitiveness. Even for the strong-performing ProPos, our method yields an average improvement of approximately 3.2%, allowing it to achieve state-of-the-art (SOTA) performance on CIFAR-10, CIFAR-20, and STL-10 while closely approaching SOTA on other datasets. Moreover, our method further pushes SOTA performance across different existing models on ImageNet-10 and ImageNet-Dogs, highlighting its robust generalizability and effectiveness. Additionally, we also provide results on the large-scale dataset Tiny-ImageNet (200 classes, 100,000 samples) in Appendix C, where our method still achieves an improvement of over 2% compared to ProPos.

Table 2: Clustering performance (%) comparisons on five datasets. The best result for each method is highlighted in **bold**, while the overall best result is marked with an underline. Average performance, standard deviation, and significance analysis are provided in Appendix C, Table 18 and Table 19.

| Method | CIFAR-10 | | | CIFAR-20 | | | STL-10 | | | ImageNet-10 | | | ImageNet-Dogs | | | |
|---|---|---|---|---|---|---|---|---|---|---|---|---|---|---|---|---|
| | NMI | ACC | ARI | NMI | ACC | ARI | NMI | ACC | ARI | NMI | ACC | ARI | NMI | ACC | ARI | Average |
| NNM [7] | 74.8 | 84.3 | 70.9 | 48.4 | 47.7 | 31.6 | 69.4 | 80.8 | 65.0 | - | - | - | - | - | - | - |
| GCC [40] | 76.4 | 85.6 | 72.8 | 47.2 | 47.2 | 30.5 | 68.4 | 78.8 | 63.1 | 84.2 | 90.1 | 82.2 | 49.0 | 52.6 | 36.2 | 64.3 |
| IDFD [34] | 71.1 | 81.5 | 66.3 | 42.6 | 42.5 | 26.4 | 64.3 | 75.6 | 57.5 | 89.8 | 95.4 | 90.1 | 54.6 | 59.1 | 41.3 | 63.9 |
| TCL [21] | 81.9 | 88.7 | 78.0 | 52.9 | 53.1 | 35.7 | 79.9 | 86.8 | 75.7 | 87.5 | 89.5 | 83.7 | 62.3 | 64.4 | 51.6 | 71.4 |
| TCC [32] | 79.0 | 90.6 | 73.3 | 47.9 | 49.1 | 31.2 | 73.2 | 81.4 | 68.9 | 84.8 | 89.7 | 82.5 | 55.4 | 59.5 | 41.7 | 67.2 |
| SPICE [27] | 85.8 | 91.7 | 83.6 | 58.3 | 58.4 | 42.2 | 86.0 | 92.9 | 85.3 | 90.2 | 95.9 | 91.2 | 62.7 | 67.5 | 52.6 | 76.3 |
| SeCu [31] | 86.1 | 93.0 | 85.7 | 55.1 | 55.2 | 39.7 | 73.3 | 83.6 | 69.3 | - | - | - | - | - | - | - |
| DPAC [38] | 87.0 | 93.4 | 86.6 | 51.2 | 55.5 | 39.3 | 86.3 | 93.4 | 86.1 | 92.5 | 97.0 | 93.5 | 66.7 | 72.6 | 59.8 | 77.4 |
| CC [20] | 76.9 | 85.2 | 72.8 | 47.7 | 41.7 | 28.8 | 73.0 | 80.0 | 68.1 | 86.0 | 89.9 | 82.3 | 65.4 | 69.6 | 56.0 | 68.2 |
| CC+Ours | **82.7** | **88.2** | **78.5** | **53.1** | **48.9** | **35.2** | **74.0** | **80.8** | **69.4** | **86.5** | **90.7** | **83.3** | **67.9** | **70.6** | **58.5** | **71.3(+3.1)** |
| SCAN [36] | 82.5 | 90.3 | 80.8 | 54.0 | 53.1 | 38.5 | 83.6 | 91.4 | 82.5 | 93.8 | 97.6 | 94.8 | 71.1 | 73.7 | 63.4 | 76.7 |
| SCAN+Ours | **84.4** | **90.8** | **82.0** | **57.0** | **54.5** | **40.6** | **84.2** | **91.7** | **83.2** | **94.2** | **97.8** | **95.2** | **73.0** | **74.5** | **65.0** | **77.9(+1.2)** |
| CDC [16] | 89.0 | 94.7 | 89.1 | 60.6 | 61.6 | 46.3 | 85.8 | 93.0 | 85.5 | 93.1 | 97.3 | 94.1 | 76.8 | 79.2 | 70.2 | 81.1 |
| CDC+Ours | **89.9** | **95.1** | **90.0** | **63.0** | **62.7** | **48.4** | **86.6** | **93.4** | **86.4** | **97.3** | **97.3** | **94.2** | **77.5** | **79.7** | **71.4** | **81.9(+0.8)** |
| BYOL [9] | 78.0 | 87.5 | 75.2 | 53.3 | 52.3 | 36.0 | 75.4 | 86.1 | 71.5 | 88.4 | 94.7 | 88.9 | 69.7 | 72.9 | 60.9 | 72.7 |
| BYOL+Ours | **85.2** | **91.5** | **83.0** | **58.1** | **54.7** | **41.5** | **80.8** | **90.2** | **79.9** | **89.9** | **95.7** | **90.8** | **73.4** | **77.1** | **67.2** | **77.3(+4.6)** |
| CoNR [39] | 86.7 | 93.2 | 86.1 | 61.7 | 59.7 | 45.0 | 85.2 | 92.6 | 84.6 | 91.1 | 96.4 | 92.2 | 74.2 | 80.2 | 67.6 | 79.8 |
| CoNR+Ours | **88.0** | **94.1** | **87.9** | **62.2** | **60.2** | **45.9** | **85.6** | **92.8** | **85.0** | **91.4** | **96.5** | **92.4** | **74.6** | **80.7** | **68.3** | **80.4(+0.6)** |
| ProPos [13] | 88.1 | 94.4 | 88.3 | 60.7 | 61.6 | 44.4 | 83.1 | 91.6 | 82.5 | 90.0 | 95.8 | 91.0 | 72.7 | 76.9 | 66.4 | 79.2 |
| ProPos+Ours | **91.1** | **96.0** | **91.6** | **64.5** | **63.9** | **49.2** | **86.7** | **93.6** | **86.6** | **92.7** | **97.1** | **93.7** | **76.3** | **79.7** | **70.7** | **82.2(+3.0)** |

To further illustrate the effectiveness of our method, we visualized learned representations using t-SNE as shown in Fig. 4. Compared to original ProPos, DCBoost produces a more globally well-structured embedding space, characterized by tighter intra-class clustering and increased inter-class separation. This improvement is particularly evident in complex datasets such as CIFAR-20 in Fig. 1(b) and ImageNet-Dogs in Fig. 4(d). To quantitatively complement these visual observations, as shown in Fig. 5. For silhouette coefficient, which serves as an indicator of global structural quality, our method significantly boosts the silhouette coefficient across various existing deep clustering models, particularly on ProPos and BYOL, indicating a more coherent global feature structure with improved inter-class separation. Meanwhile, the $k$-NN accuracy, which measures local consistency, remains high and even shows a slight improvement, demonstrating that good local structures are not only preserved but also further refined. These observations highlight the effectiveness of leveraging reliable local structural cues to guide and improve the global structure.

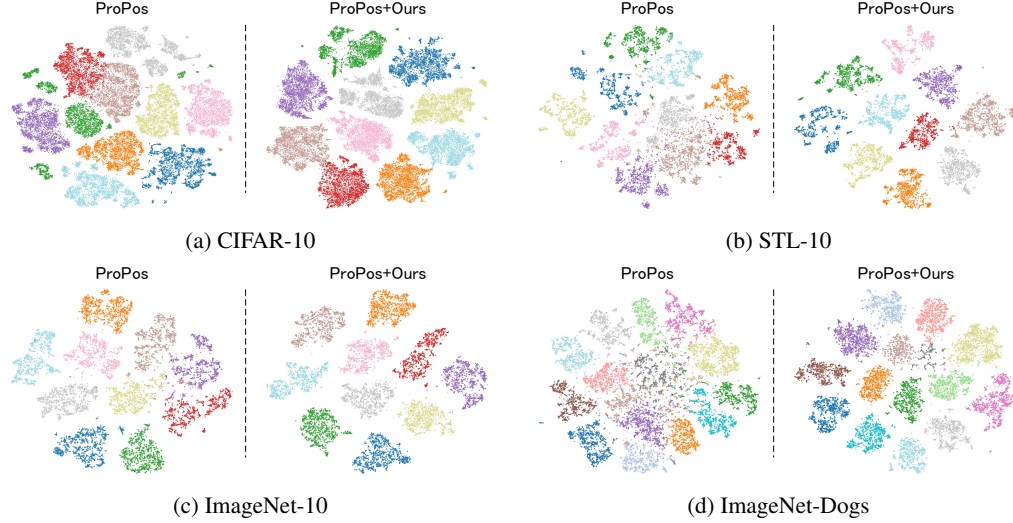

(a) CIFAR-10

(b) STL-10

(c) ImageNet-10

(d) ImageNet-Dogs

Figure 4: T-SNE visualization of ProPos (left) and ProPos+Ours (right) on four datasets. The visualization of CIFAR-20 has been presented on Fig. 1(b).

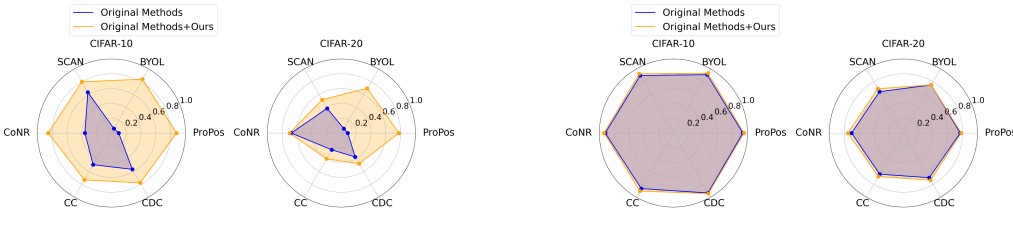

(a) Silhouette Coefficient          (b) $k$-NN Accuracy

Figure 5: Influence on silhouette coefficient (global structure) and $k$-NN accuracy (local structure).

Table 3: Ablation study (%) on CIFAR-10, CIFAR-20 and STL-10.

| Filter | $L_{ins}$ | $L_{pos}$ | $L_{neg}$ | | CIFAR-10 | | | CIFAR-20 | | | STL-10 | | | |
|---|---|---|---|---|---|---|---|---|---|---|---|---|---|---|
| | | | | | NMI | ACC | ARI | NMI | ACC | ARI | NMI | ACC | ARI | Average |
| - | - | - | - | ProPos | 88.1 | 94.4 | 88.3 | 60.7 | 61.6 | 44.4 | 83.1 | 91.6 | 82.5 | 77.2 |
| × | ✓ | ✓ | × | | 88.0 | 94.2 | 87.9 | 59.5 | 57.7 | 39.6 | 83.9 | 90.6 | 79.8 | 75.7 |
| ✓ | ✓ | ✓ | × | | 90.5 | 95.6 | 90.8 | 62.4 | 61.8 | 46.5 | 84.8 | 92.2 | 83.9 | 78.7 |
| ✓ | ✓ | × | ✓ | | 88.3 | 94.4 | 88.3 | 57.1 | 59.1 | 42.1 | 83.6 | 92.0 | 83.4 | 76.5 |
| ✓ | × | ✓ | ✓ | | 83.1 | 91.1 | 81.5 | 63.1 | 62.5 | 46.9 | 11.6 | 20.5 | 6.7 | 51.9 |
| × | ✓ | ✓ | ✓ | | 89.0 | 94.9 | 89.1 | 63.1 | 62.5 | 47.0 | 84.3 | 92.0 | 83.1 | 78.3 |
| ✓ | ✓ | ✓ | ✓ | ProPos+Ours | **91.1** | **96.0** | **91.6** | **64.5** | **63.9** | **49.2** | **86.7** | **93.6** | **86.6** | **80.4** |

## 4.3 Ablation Study

**Each loss term contribute to the clustering performance improvement.** From Table 3, we observe that incorporating all proposed losses while keeping our sample selection mechanism fixed yields the best clustering performance. Removing any individual loss component results in a performance drop, highlighting their complementary effectiveness. Specifically, removing the instance consistency loss $L_{ins}$ eliminates instance-level consistency constraints. The model struggles to preserve meaningful structure, leading to a collapse in clustering performance, as evidenced by obvious ACC degradation on both CIFAR-10 (96.0% → 91.1%) and STL-10 (93.6% → 20.5%). Without the positive loss $L_{pos}$, intra-class consistency is not enforced. Relying solely on inter-class separation provides minimal training guidance, leading to nearly no performance improvement. The removal of negative loss $L_{neg}$ still has a noticeable impact, and its effect would be even more pronounced without our sample selection mechanism, which will be further explored in the corresponding ablation study.

**Validation of adaptive $k$-NN filtering.** Furthermore, applying all losses without sample selection significantly degrades average performance (80.4% → 78.3%), highlighting the critical importance of selecting high-confidence samples. In the absence of sample selection, the removal of $L_{neg}$ leads to partial clustering collapse, causing performance to drop below baseline. In contrast, incorporating our sample selection strategy significantly improved average performance (75.7% → 78.7%), further validating its effectiveness. As shown in Fig. 6(a), high-confidence samples consistently maintain higher accuracy than the overall dataset throughout training, highlighting their reliability in guiding the clustering process and enhancing performance. Additionally, Figs. 6(b)(c) demonstrate that our method adaptively selects the appropriate $k$ value for different datasets: achieve an ACC of 96.0% compared to 95.9% with manually set $k = 30$, and on CIFAR-20, 63.9% compared to 63.6% with $k = 10$. Beyond achieving performance gains, the key advantage is robustness—manual selection is sensitive to the chosen $k$, and inappropriate settings can cause notable performance degradation. In contrast, our adaptive strategy yields stable improvements without hyperparameter tuning.

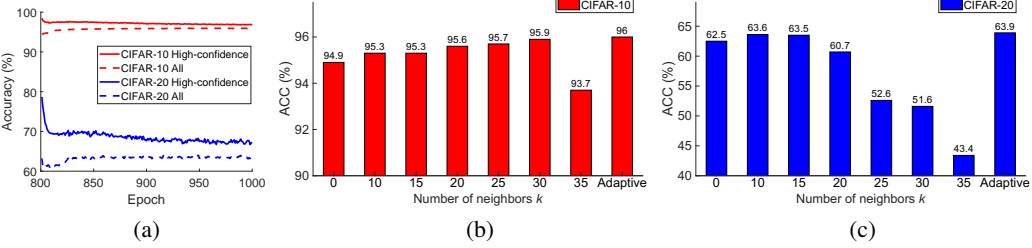

(a)              (b)              (c)

Figure 6: (**a**) Comparison of accuracy between high-confidence samples selected by our methods and all samples. Impact of the value of $k$ on clustering ACC across (**b**) CIFAR-10 and (**c**) CIFAR-20.

**Class-balanced weight $w$ in $L_{pos}$.** To relief the imbalance class distribution of the selected high-confidence samples, in Eq. 4, we introduce a weighting strategy that assigns each class an equal contribution. We conducted an ablation study with three variants: $w_0$, which disables the weighting mechanism; $w_1$, which applies the weights without enabling gradient flow; and $w_{ours}$, our complete implementation that incorporates weights into the gradient updates. The results in Table 4 show that $w_0$ yields the worst performance, as treating all high-confidence samples equally ignores class imbalance, causing classes with a larger number of high-confidence samples to dominate the training process. However, $w_1$ significantly improves performance by ensuring each class contributes equally to the objective, effectively

Table 4: Impact of class-balanced weighting on clustering performance (%).

| Method | CIFAR-10 | | | CIFAR-20 | | |
|---|---|---|---|---|---|---|
| | NMI | ACC | ARI | NMI | ACC | ARI |
| ProPos | 88.1 | 94.4 | 88.3 | 60.7 | 61.6 | 44.4 |
| $w_0$ | 90.2 | 95.3 | 90.2 | 61.9 | 61.9 | 45.7 |
| $w_1$ | 91.0 | 96.0 | 91.5 | 64.0 | 63.8 | 48.9 |
| $w_{ours}$ | **91.1** | **96.0** | **91.6** | **64.5** | **63.9** | **49.2** |

mitigating the imbalance. Finally, $w_{ours}$ achieves slightly better results than $w_1$, suggesting that the magnitude of intra-class representations through gradient-based learning of the weight modulation further refines cluster compactness and improves structural consistency. These results confirm the effectiveness of our class-balanced weighting strategy. More ablation studies are shown in the Appendix B.

## 4.4 More Discussions

**Comparison between representation-based and clustering-head-based architectures.** Among the various models we use, CC [20] incorporates both a clustering head and a representation head. To assess the impact of our method on different network components, we applied it separately to representation and clustering head, with results shown in Table 5. Across three datasets, in terms of ACC, the representation shows an increase of (3.0%, 6.1%, 1.0%), while the clustering head improves by (5.0%, 8.0%, 2.4%). The larger gains on representation indicate our method is especially effective in refining representation-based models, i.e., the representation space retains richer semantic information, enabling our method to enhance the learned features more effectively.

Table 5: Performance (%) gain comparison between clustering head (Clu) and representation (Rep).

| Method | CIFAR-10 | | | CIFAR-20 | | | ImageNet-Dogs | | |
|---|---|---|---|---|---|---|---|---|---|
| | NMI | ACC | ARI | NMI | ACC | ARI | NMI | ACC | ARI |
| CC(Clu) | 76.9 | 85.2 | 72.8 | 47.1 | 42.4 | 28.4 | 65.4 | 69.6 | 56.0 |
| CC(Clu)+Ours | 82.7(↑5.8) | 88.2(↑3.0) | 78.5(↑5.7) | 52.5(↑5.4) | 48.5(↑6.1) | 34.4(↑6.0) | 67.9(↑2.5) | 70.6(↑1.0) | 58.5(↑2.5) |
| CC(Rep) | 78.5 | 86.3 | 74.9 | 50.4 | 48.9 | 33.2 | 63.3 | 66.3 | 52.7 |
| CC(Rep)+Ours | 85.9(↑7.4) | 91.3(↑5.0) | 83.4(↑8.5) | 58.8(↑8.4) | 56.9(↑8.0) | 42.5(↑9.3) | 66.2(↑3.9) | 68.7(↑2.4) | 55.5(↑2.8) |

Table 6: Universality comparison (%) with CoNR.

| Method | CIFAR-10 | | | CIFAR-20 | | |
|---|---|---|---|---|---|---|
| | NMI | ACC | ARI | NMI | ACC | ARI |
| Propos | 88.1 | 94.4 | 88.3 | 60.7 | 61.6 | 44.4 |
| Propos+CoNR | 90.3 | 95.6 | 90.6 | 64.0 | 63.1 | 48.4 |
| Propos+Ours | **91.1** | **96.0** | **91.6** | **64.5** | **63.9** | **49.2** |
| CC | 78.5 | 86.3 | 74.9 | 50.4 | 48.9 | 33.2 |
| CC+CoNR | 84.8 | 90.8 | 82.4 | 58.4 | 55.7 | 41.4 |
| CC+Ours | **85.9** | **91.3** | **83.4** | **58.8** | **56.9** | **42.5** |

Table 7: Cluster performance (%) comparisons on CLIP-based models.

| Method | CIFAR-10 | | | CIFAR-20 | | |
|---|---|---|---|---|---|---|
| | NMI | ACC | ARI | NMI | ACC | ARI |
| SIC | 85.0 | 92.8 | 84.9 | 58.2 | 57.4 | 43.7 |
| SIC+Ours | **85.3** | **92.8** | **85.0** | **59.3** | **58.4** | **44.9** |
| TAC | 83.4 | 92.0 | 83.5 | 60.8 | 61.5 | 46.2 |
| TAC+Ours | **85.5** | **93.1** | **85.6** | **62.4** | **62.9** | **48.4** |

**Comparison with CoNR in terms of universality.** CoNR [39] enhances intra-class compactness by leveraging contextual cues and filtering boundary samples, which shares some similarities with our method. Therefore, we integrated CoNR into ProPos [13] and CC [20] to evaluate its universality and compare its effectiveness with our method. The results in Table 6 indicate that although CoNR improves performance, the gains are consistently smaller than those of our method. This is largely due to the fact that CoNR limited use of class-level information, as it mainly enforces local consistency with a small set of positive pairs. In contrast, our method can more efficiently exploit reliable local structural information to guide the refinement of global feature organization, leading to more substantial improvements in clustering performance.

Table 8: Influence of different sample selection methods on clustering performance(%).

| Method | CIFAR-10 | | | CIFAR-20 | | | ImageNet-Dogs | | |
|---|---|---|---|---|---|---|---|---|---|
| | NMI | ACC | ARI | NMI | ACC | ARI | NMI | ACC | ARI |
| ProPos | 88.1 | 94.4 | 88.3 | 60.7 | 61.6 | 44.4 | 72.7 | 76.9 | 66.4 |
| ProPos+Ours (MOIT sel.) | 90.7 | 95.8 | 91.1 | 64.1 | 63.2 | 47.8 | 73.5 | 77.1 | 66.6 |
| ProPos+Ours (SSR sel.) | 90.9 | 95.9 | 91.3 | 63.9 | 63.6 | 48.3 | 75.3 | 79.0 | 70.1 |
| ProPos+Ours | **91.1** | **96.0** | **91.6** | **64.5** | **63.9** | **49.2** | **76.3** | **79.7** | **70.7** |

**Extension to CLIP-based deep clustering models.** We further applied our method to CLIP-based models SIC [1] and TAC [22], where clustering results are obtained using an image encoder followed by a clustering head. We incorporated our method into their frameworks by leveraging the existing branch and fine-tuning the clustering head. We re-implemented experiments on the merged dataset (Train+Test), while keeping other settings consistent with [1, 22]. As shown in Table 7, our method brings consistent gains, confirming its effectiveness even in CLIP-based deep clustering models.

**Comparison with different high-confidence sample selection methods.** To better understand the effect of different sample selection mechanisms, we compare our adaptive $k$-NN approach with two representative noisy-label filtering methods: MOIT [28] and SSR [8]. MOIT detects noisy labels by measuring disagreement between a sample's annotated label and the class distribution of its $k$-nearest neighbors in the global feature space, while SSR relies on label–neighborhood consistency. Both approaches require global $k$-NN search over the entire dataset with a fixed $k$, and involve additional hyperparameters such as consistency thresholds and confidence scores.

In contrast, our method is parameter-free and applies an adaptive $k$-NN search locally within each mini-batch, thereby avoiding the overhead of global retrieval. We re-implemented MOIT and SSR sample selection strategies and applied them on ProPos, replacing only the selection module while keeping all other training settings unchanged. As shown in the Table 8, integrating MOIT or SSR filtering into ProPos yields consistent improvements over the baseline, confirming the importance of neighborhood-based selection. However, our adaptive mini-batch selection further achieves the best performance across all datasets and metrics, without introducing extra hyperparameters or global $k$-NN computations. Additionally, when their selection is applied using their own loss to train the model, performance does not improve as shown in 9. More experiments are shown in the Appendix C.

Table 9: CIFAR-10 ACC (%) using MOIT and SSR as complete methods.

| Method | ACC |
|---|---|
| ProPos | 94.4 |
| ProPos+MOIT | 94.4 |
| ProPos+SSR | 92.5 |
| ProPos+Ours | **96.0** |

## 5 Conclusion

In this paper, we have presented **DCBoost**, a universal, parameter-free plug-and-play method to improve existing deep clustering models. Motivated by the observation that existing methods often fail to learn reliable global structures despite consistent local patterns, DCBoost adaptively selects high-confidence samples via local $k$-NN consistency to guide intra-class compactness and inter-class separation, leading to a better global structure and accordingly higher clustering performance. Extensive results show consistent improvements across different deep clustering models and benchmarks. A potential limitation is that our method assumes the number of clusters is known and lacks mechanisms to handle highly imbalanced or non-uniform data distributions, which may reduce its adaptability in complex real-world scenarios. In the future, we plan to scale DCBoost to large-scale datasets, explore more adaptive selection strategies and multimodal extensions to overcome its current limitations and further boost clustering performance.

**Acknowledgments**

This work was supported in part by the National Natural Science Foundation of China under Grants U24A20322, 62576094 and 62422118, in part by the Hong Kong UGC under grants UGC/FDS11/E03/24, UGC/FDS11/E03/25, and in part by the Hong Kong Research Grants Council

under Grant 11219324. This research work was also supported by the Big Data Computing Center of Southeast University.

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

# Appendix

## A  More Implementation Details

**Dataset, image size and backbone.** Following [20, 13, 39, 16], we used the merged training and testing sets for CIFAR-10 and CIFAR-20, only the training set for ImageNet-10 and ImageNet-Dogs. Since our method improved upon existing models, for STL-10, we no longer required unlabeled samples for pre-training and instead used the merged training and testing set. For all methods except BYOL [9] and CoNR [39], image sizes were set as follows: 32×32 for CIFAR-10 and CIFAR-20, 96×96 for STL-10, and 224×224 for ImageNet-10 and ImageNet-Dogs, following [13, 16]. ResNet-34 served as the backbone for all these methods. For BYOL and CoNR, slight modifications were made to ensure a fair comparison. Specifically, ResNet-18 was used for CIFAR-10 and CIFAR-20, while ImageNet-10 adopts a 96×96 image size, with all other settings unchanged, following [39]. For all methods on CIFAR-10 and CIFAR-20, we followed [13, 39, 16] by replacing the first convolutional filter (7×7, stride 2) with a 3×3 filter (stride 1) and removing the first max-pooling layer to better accommodate the smaller image resolution.

**Model adaptation.** To integrate various clustering models into our framework, we adopted a universal adaptation strategy. Specifically, we retained the entire network originally used for clustering—whether it comprises a backbone followed by a clustering head or a backbone followed by a projector—as the target network $f_t(\cdot)$ and duplicated it to form the online network $f_o(\cdot)$. Additionally, we introduced a randomly initialized predictor subsequent to the online network, establishing an asymmetric architecture similar to BYOL [9]. For sample selection, We only use the target network's features for $k$-NN retrieval ($S_{ij} = \cos\left(z_i^t, z_j^t\right)$, where $S$ is similarity matrix). This is because the predictor followed by the online network is randomly initialized, and may introduce noise or unstable representations early in training. To ensure reliable consistency estimation, we rely solely on the more stable target encoder output in such cases. We first conducted warm-up training using $L_{ins}$ for 10 epochs. This step helped the predictor develop initial feature-capturing capability, mitigating the instability that a randomly initialized predictor might introduce and ensuring smooth convergence[1].

**Experiment settings.** We strictly followed the data augmentation protocols from [13, 39], which applied ResizedCrop, ColorJitter, Grayscale, and HorizontalFlip to all datasets, and additionally applied GaussianBlur specifically to 224×224 images from ImageNet-10 and ImageNet-Dogs. We adopted the stochastic gradient descent (SGD) optimizer, the base learning rate was 0.05, scaled linearly with the batch size of 256. The learning rates for the predictor were 10× as the learning rate of feature models. For the exponential moving average hyperparameter to update the online network, we set it to 0.996. All these settings strictly followed [13]. We inserted all the existing models into our model at the 800th epoch to inherit the hyper-parameters at that time and continued to train another 200 epochs.

**Evaluation metrics.** We employed three commonly used clustering metrics to evaluate performance: Normalized Mutual Information (NMI) [33], Accuracy (ACC) [19], and Adjusted Rand Index (ARI) [14]. Higher scores indicate better clustering performance.

**Baseline methods.** For a fair comparison, we strictly followed the experimental settings of [16] to re-implement CC [20], SCAN [36], and CDC [16]. The predictor subsequent to the online network consists of an MLP with the structure ($h$D-BN [15]-ReLU [26]-$d$D), where $h$=512 was set to match the hidden layer dimension of the projector. The output dimension $d$ remained the same as the input dimension of predictor to ensure consistency. And we also rigorously adhered to [13, 39] when re-implementing BYOL [9], CoNR [39], and ProPos [13]. For other deep clustering models, including NNM [7], GCC [40], IDFD [34], TCL [21], TCC [32], SPICE [27], SeCu [31], and DPAC [38], we directly cited the results they reported.

---

[1]For models already based on BYOL-like architecture, the required architecture is inherently present. Therefore, our method could be directly applied without architectural changes or additional initialization strategies. We also combine outputs from both the online and target networks for sample selection, because the hybrid approach balances the stability of the target network, and the real-time adaptability of the online network.

# B  More Ablation Study

**Robustness to cluster number.** Previous experiments assumed prior knowledge of the true number of categories, which is often unavailable in real-world applications. To evaluate the robustness of our method to different clustering numbers, we conducted experiments on CIFAR-20 with varying numbers of clusters ($c = 10, 20, 30, 40, 50$), simulating both underclustering and overclustering scenarios. We applied $k$-means and evaluate clustering performance under the different predefined $c$. As shown in the Table 10, both NMI and ARI show clear improvements compared to the baseline clustering setting. This suggests that our method enhances the model's ability to discover meaningful partitions even when the number of clusters deviates from the ground-truth.

Table 10: Clustering performance (%) on CIFAR-20 with varying cluster numbers.

| | CIFAR-20 | | | | | | | | | |
|---|---|---|---|---|---|---|---|---|---|---|
| Class Number | $c = 10$ | | $c = 20$ | | $c = 30$ | | $c = 40$ | | $c = 50$ | |
| Metric | NMI | ARI | NMI | ARI | NMI | ARI | NMI | ARI | NMI | ARI |
| ProPos | 53.6 | 32.4 | 60.7 | 44.4 | 59.5 | 38.2 | 59.5 | 36.6 | 58.8 | 33.3 |
| ProPos+Ours | **56.1** | **34.4** | **64.5** | **49.2** | **62.1** | **44.6** | **62.7** | **42.3** | **62.2** | **38.1** |

**Robustness to pseudo-label generation strategies.** During training, pseudo labels are typically obtained by performing $k$-means clustering of all samples at each epoch. For models equipped with a clustering head, another option is to directly generate pseudo labels from the softmax outputs, since the output dimension corresponds to the number of clusters. In this study, we explored the impact of different pseudo-labeling strategies on clustering performance. As shown in Table 11 , our proposed method consistently improves clustering performance across various pseudo-label generation strategies, demonstrating its general applicability.

Table 11: Influence on model performance (%) with different pseudo-label generation strategies.

| Method | CIFAR-20 | | | CIFAR-10 | | |
|---|---|---|---|---|---|---|
| | NMI | ACC | ARI | NMI | ACC | ARI |
| CDC | 60.6 | 61.6 | 46.3 | 89.0 | 94.7 | 89.1 |
| CDC+Ours ($k$-means) | **63.1** | 62.6 | **48.6** | **89.9** | **95.1** | **90.0** |
| CDC+Ours (softmax) | 62.6 | **63.2** | 48.3 | **89.9** | **95.1** | **90.0** |

**NMI and ARI on different $k$ selecting strategies.** As shown in Fig. 7, consistent with the previous trend of ACC in Fig. 6, the $k$ value selected manually varies between different datasets and may degrade cluster performance if set inappropriately, while the proposed adaptive $k$ selection strategy effectively addresses this issue.

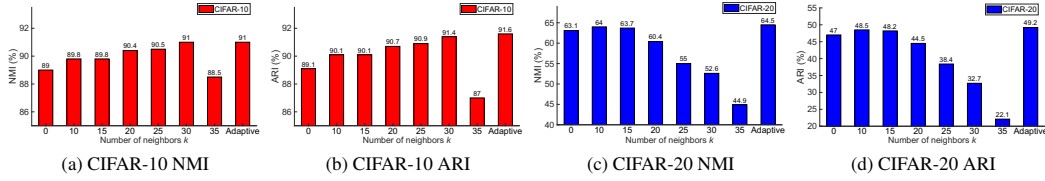

(a) CIFAR-10 NMI     (b) CIFAR-10 ARI     (c) CIFAR-20 NMI     (d) CIFAR-20 ARI

Figure 7: Impact of neighbor number $k$ on NMI and ARI across different datasets.

**The dynamic retrieval parameter $m$ setting.** In our method, different $k$ values are evaluated to select an appropriate $k$, and here we conduct an ablation study on the efficiency of this procedure. Although multiple candidate $k$ values within $[1, m]$ are considered, the $m$-nearest neighbors are computed only once per batch (e.g., $m = 50$), and all scores under different $k$ are derived from this single retrieval. The operation performs label consistency checks and determines the favorable $k$

using efficient matrix operations. As a result, with a batch size $B$, the computational complexity is $O(B \cdot m)$, and the overall training cost is not significantly affected by the choice of $m$.

Further experiments on CIFAR-10 show that varying $m$ (e.g., $m = 25, 50, 75$), using a fixed $k = 25$, or even removing the $k$-NN filtering altogether (i.e., $k = 0$, using all samples) leads to negligible differences in runtime. Meanwhile, the dynamic strategy consistently improves sample selection while introducing no observable overhead (see Table 12).

Table 12: Training time for different searching space.

| Method | NMI | ACC | ARI | Time (h:m:s) |
|---|---|---|---|---|
| $k = 0$ | 89.1 | 94.9 | 89.3 | 4:22:32 |
| $k = 25$ | 91.1 | 91.6 | 96.0 | 4:25:58 |
| $m = 25$ | 91.2 | 91.7 | 96.1 | 4:25:51 |
| $m = 50$ | 91.1 | 91.6 | 96.0 | 4:29:00 |
| $m = 75$ | 91.1 | 91.6 | 96.0 | 4:25:58 |

## C  More Experiments

**Results on large-scale datasets.** We further evaluated our method on the large-scale Tiny-ImageNet dataset, which contains 200 classes and 100,000 training images. We followed settings in [16] with a ResNet-34 backbone and a image size of 64×64. Due to memory constraints, the batch size cannot be set very large, resulting in only a few same-class samples per batch, limiting the effectiveness of our batch-wise adaptive $k$-NN selection. To mitigate this, we adopt a queue mechanism to expand the neighborhood search space, following [13], and manually set a fixed $k = 5$ to preserve high-confidence sample selection. Experimental results in Table 13 demonstrate that our method is not only effective on standard datasets but also adaptable to larger-scale datasets.

Table 13: Clustering performance (%) on Tiny-ImageNet.

| Method | Tiny-ImageNet | | |
|---|---|---|---|
| | NMI | ACC | ARI |
| ProPos | 46.5 | 29.8 | 18.2 |
| ProPos+Ours | **48.9** | **31.9** | **19.8** |
| CDC | 47.5 | 33.9 | 19.9 |
| CDC+Ours | **48.0** | **34.7** | **20.9** |

**Scalability to datasets with many categories.** To evaluate the scalability of our method to scenarios with a large number of categories, we further conducted experiments on CIFAR-100 (100 classes), Tiny-ImageNet (200 classes), and ImageNet-1K (1000 classes). As shown in the Table 14, our method consistently improves over the ProPos baseline across all datasets. These results confirm that our method is scalable and effective even as the number of clusters increases from tens to hundreds or thousands.

Table 14: Clustering performance(%) on datasets with many categories.

| Method | CIFAR-100 | | | Tiny-ImageNet | | | ImageNet-1K | | |
|---|---|---|---|---|---|---|---|---|---|
| | NMI | ACC | ARI | NMI | ACC | ARI | NMI | ACC | ARI |
| ProPos | 65.7 | 55.2 | 40.1 | 46.5 | 29.8 | 18.2 | 51.3 | 22.0 | 13.4 |
| ProPos+Ours | **67.4** | **57.1** | **43.4** | **48.9** | **31.9** | **19.8** | **54.5** | **23.2** | **15.1** |

**Results on long-tailed datasets.** To further validate the robustness of our method, we conducted additional experiments on long-tailed CIFAR-10 and CIFAR-20 with an imbalance ratio of 10, where the number of training samples per class follows an exponential decay controlled by a ratio between

the number of samples in the most frequent class and the least frequent class. For instance, with ratio=10, the head class contains 5000 samples while the tail class has only 500. As shown in Table 15, our approach consistently improves clustering performance over strong baselines. These gains highlight that the proposed $k$-NN-based selection strategy remains effective and robust, even under class-imbalanced conditions. Notably, although our method does not explicitly incorporate mechanisms tailored for non-uniform data distributions, it still yields consistent improvements across different datasets and baselines.

Table 15: Clustering performance(%) on long-tailed datasets.

| Method | CIFAR10-LT | | | CIFAR20-LT | | |
|---|---|---|---|---|---|---|
| | NMI | ACC | ARI | NMI | ACC | ARI |
| ProPos | 57.1 | 48.3 | 40.7 | 47.6 | 42.1 | 29.9 |
| ProPos+Ours | **61.3** | **49.9** | **43.3** | **49.1** | **43.4** | **31.5** |
| CoNR | 67.2 | 64.6 | 56.7 | 51.2 | 43.9 | 30.3 |
| CoNR+Ours | **68.4** | **65.3** | **59.1** | **51.4** | **44.1** | **33.3** |
| LFSS [24] | 57.9 | 56.2 | 43.0 | 46.7 | 41.4 | 28.3 |
| LFSS+Ours | **61.3** | **60.1** | **46.9** | **47.5** | **42.4** | **29.4** |

**Clustering performance curve comparison.** To assess the observed performance gains are derived from our proposed method rather than prolonged training, we conducted a study comparing the original model with and without our method under extended training epochs. As shown in Fig. 8, simply extending the training time of ProPos does not lead to significant improvement and may even degrade performance. In contrast, incorporating our method results in a notable and stable enhancement in clustering ACC.

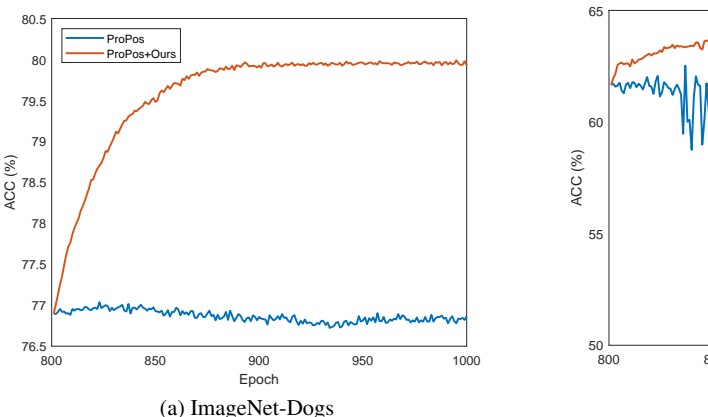
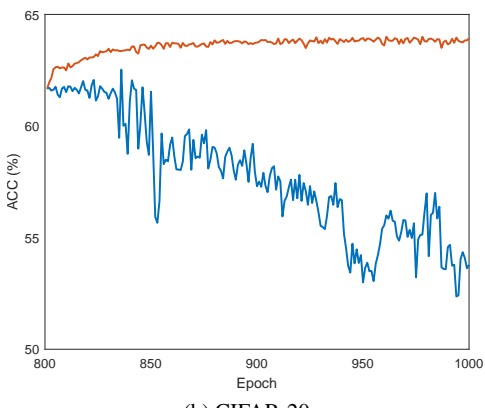

(a) ImageNet-Dogs    (b) CIFAR-20

Figure 8: Evaluating the effect of prolonged training and our method.

**Training efficiency.** We reported the running time per epoch on a single RTX3090 GPU in Table 16. Compared to CDC [16] (100 epochs after pretraining), ProPos (200 epochs), and SCAN [36] (300 epochs after pretraining), our method exhibits a comparable per-epoch runtime to ProPos and SCAN, and is significantly faster than CDC. This suggests that our framework introduces negligible computational overhead and maintains high efficiency during training.

**Enhancement on CoNR.** CoNR [39] can be applied to existing models to boost their performance. However, when our method is further applied on top of CoNR-enhanced models, we observe additional performance gains in Table 17. This demonstrates that our method complements CoNR's ability to improve clustering performance.

**Average performance and standard deviation for deep clustering models.** To ensure the reliability of our experimental results, we conducted five runs across various models. Table 18 reports the average performance along with standard deviations. For each existing deep clustering model,

Table 16: Running time comparison.

| Running Time Per Epoch (Minute) | CIFAR-10 | CIFAR-20 | STL-10 | ImageNet-10 | ImageNet-Dogs |
|---|---|---|---|---|---|
| CDC | 5.2 | 1.4 | 0.8 | 3.6 | 3.5 |
| SCAN | 0.9 | 0.9 | 0.3 | 0.8 | 1.1 |
| ProPos | 1.2 | 1.1 | 0.4 | 1.4 | 2.0 |
| CDC/SCAN+Ours (200 epochs) | 1.1 | 1.1 | 0.3 | 1.4 | 1.7 |
| ProPos+Ours (200 epochs) | 1.2 | 1.1 | 0.4 | 1.4 | 1.8 |

Table 17: Further enhancement (%) after enhancement of CoNR on ProPos and CC.

| Method | CIFAR-10 | | | CIFAR-20 | | |
|---|---|---|---|---|---|---|
| | NMI | ACC | ARI | NMI | ACC | ARI |
| ProPos | 88.1 | 94.4 | 88.3 | 60.7 | 61.6 | 44.4 |
| ProPos+CoNR | 90.3 | 95.6 | 90.6 | 64.0 | 63.1 | 48.4 |
| ProPos+CoNR+Ours | **91.1** | **96.0** | **91.5** | **64.2** | **63.5** | **48.8** |
| CC | 78.5 | 86.3 | 74.9 | 50.4 | 48.9 | 33.2 |
| CC+CoNR | 84.8 | 90.8 | 82.4 | 58.4 | 55.7 | 41.4 |
| CC+CoNR+Ours | **86.9** | **92.1** | **84.7** | **58.8** | **56.0** | **42.2** |

integrating our method consistently leads to notable performance gains compared to the original results, **all of which are statistically significant under a 5% $t$-test**, corresponding $p$-values are shown in Table 19.

Table 18: Clustering performance NMI, ACC, ARI (mean±std %) of different deep clustering models on five image benchmarks.

| Method | CIFAR-10 | | | CIFAR-20 | | | STL-10 | | | ImageNet-10 | | | ImageNet-Dogs | | |
|---|---|---|---|---|---|---|---|---|---|---|---|---|---|---|---|
| | NMI | ACC | ARI | NMI | ACC | ARI | NMI | ACC | ARI | NMI | ACC | ARI | NMI | ACC | ARI |
| CC | 74.9±3.1 | 82.3±4.4 | 69.3±4.5 | 48.2±0.6 | 43.4±2.3 | 30.3±1.4 | 76.9±3.7 | 85.0±4.8 | 73.5±5.5 | 86.8±1.0 | 90.4±0.7 | 83.4±1.6 | 63.2±1.9 | 67.0±2.8 | 52.6±2.9 |
| CC+Ours | 81.2±2.4 | 86.5±3.2 | 76.5±3.2 | 54.3±1.3 | 50.5±1.5 | 36.9±1.6 | 78.2±4.3 | 85.9±5.0 | 75.1±6.1 | 87.1±0.9 | 90.9±0.3 | 84.4±1.1 | 67.3±0.6 | 69.7±2.1 | 57.7±0.8 |
| SCAN | 84.1±2.1 | 91.4±1.7 | 82.9±3.1 | 53.7±1.6 | 51.9±2.0 | 37.3±2.7 | 84.0±0.4 | 91.8±0.3 | 83.3±0.7 | 90.6±2.8 | 93.4±3.7 | 88.9±5.1 | 70.5±3.6 | 72.6±6.0 | 62.5±5.8 |
| SCAN+Ours | 85.9±2.1 | 91.9±1.7 | 84.1±3.2 | 56.4±1.9 | 53.5±1.8 | 39.4±2.7 | 84.8±0.6 | 92.1±0.4 | 84.1±0.8 | 91.1±2.7 | 93.5±5.7 | 89.2±5.2 | 73.3±3.3 | 74.5±6.5 | 65.3±5.6 |
| CDC | 88.0±1.0 | 93.7±0.9 | 87.3±1.6 | 60.4±0.7 | 60.5±1.7 | 45.7±1.2 | 86.1±0.4 | 93.1±0.1 | 85.8±0.4 | 92.8±0.4 | 97.2±0.2 | 93.8±0.5 | 75.3±2.0 | 77.6±2.6 | 68.2±2.8 |
| CDC+Ours | 89.0±0.9 | 94.3±0.9 | 88.5±1.6 | 62.6±0.9 | 61.7±1.9 | 47.7±1.6 | 86.6±0.2 | 93.4±0.1 | 86.3±0.2 | 93.1±0.4 | 97.2±0.1 | 94.0±0.3 | 76.2±1.8 | 78.3±2.5 | 69.6±2.7 |
| BYOL | 76.7±2.6 | 86.6±1.8 | 73.6±3.3 | 53.5±0.5 | 51.8±0.9 | 35.6±0.9 | 74.5±1.9 | 85.0±2.4 | 70.2±2.8 | 87.6±1.6 | 94.3±0.8 | 88.1±1.7 | 69.6±0.3 | 72.8±0.3 | 61.0±0.1 |
| BYOL+Ours | 84.7±0.8 | 91.1±1.1 | 82.2±1.8 | 57.8±0.7 | 54.9±1.8 | 41.1±1.0 | 81.3±2.8 | 89.4±3.5 | 79.1±5.4 | 89.9±0.2 | 95.7±0.1 | 90.8±0.2 | 74.2±0.6 | 77.4±0.5 | 68.0±0.7 |
| CoNR | 86.3±0.8 | 92.6±1.0 | 85.2±1.6 | 61.2±0.9 | 59.5±0.4 | 44.9±0.1 | 83.8±1.6 | 91.7±1.1 | 82.7±2.2 | 90.8±0.2 | 96.3±0.2 | 91.7±0.4 | 73.7±0.6 | 78.2±2.3 | 66.5±1.3 |
| CoNR+Ours | 87.5±1.0 | 93.6±1.0 | 86.9±1.8 | 61.8±0.8 | 60.3±0.4 | 46.0±0.5 | 84.9±0.8 | 92.3±0.6 | 84.1±1.0 | 91.3±0.1 | 96.4±0.0 | 92.3±0.1 | 74.9±0.3 | 79.5±1.4 | 68.6±0.3 |
| ProPos | 88.4±0.6 | 94.6±0.3 | 88.7±0.6 | 60.3±0.5 | 60.0±1.5 | 44.3±0.5 | 82.4±1.8 | 91.1±1.2 | 81.6±2.2 | 88.5±1.8 | 95.0±0.9 | 89.4±1.9 | 72.9±0.1 | 76.8±0.1 | 66.3±0.3 |
| ProPos+Ours | 90.2±0.6 | 95.5±0.3 | 90.5±0.7 | 63.1±1.0 | 62.1±1.4 | 47.6±1.1 | 86.3±0.5 | 93.3±0.3 | 85.9±0.7 | 91.7±1.1 | 96.6±0.5 | 92.7±1.1 | 76.1±0.7 | 79.8±0.1 | 70.5±0.5 |

Table 19: T-test results comparing baseline deep clustering models and their enhanced versions with our method across five image benchmarks. A $p$-value less than 0.05 indicates a statistically significant improvement over the baseline.

| Method | CIFAR-10 | | | CIFAR-20 | | | STL-10 | | | ImageNet-10 | | | ImageNet-Dogs | | |
|---|---|---|---|---|---|---|---|---|---|---|---|---|---|---|---|
| | NMI | ACC | ARI | NMI | ACC | ARI | NMI | ACC | ARI | NMI | ACC | ARI | NMI | ACC | ARI |
| CC+Ours | 0.003 | 0.013 | 0.005 | 0.002 | 0.004 | 0.001 | 0.046 | 0.025 | 0.040 | 0.007 | 0.027 | 0.017 | 0.018 | 0.043 | 0.031 |
| SCAN+Ours | 0.001 | 0.002 | 0.003 | 0.003 | 0.004 | 0.000 | 0.001 | 0.016 | 0.000 | 0.003 | 0.019 | 0.018 | 0.015 | 0.048 | 0.023 |
| CDC+Ours | 0.001 | 0.019 | 0.014 | 0.000 | 0.000 | 0.000 | 0.002 | 0.001 | 0.001 | 0.037 | 0.001 | 0.047 | 0.000 | 0.001 | 0.000 |
| BYOL+Ours | 0.002 | 0.001 | 0.001 | 0.002 | 0.005 | 0.000 | 0.000 | 0.003 | 0.004 | 0.028 | 0.018 | 0.020 | 0.000 | 0.000 | 0.000 |
| CoNR+Ours | 0.001 | 0.002 | 0.003 | 0.007 | 0.021 | 0.020 | 0.036 | 0.044 | 0.047 | 0.017 | 0.032 | 0.020 | 0.040 | 0.032 | 0.040 |
| ProPos+Ours | 0.003 | 0.004 | 0.004 | 0.004 | 0.037 | 0.002 | 0.020 | 0.029 | 0.022 | 0.008 | 0.014 | 0.010 | 0.008 | 0.001 | 0.004 |

**Visualizations of adaptive $k$ during training.** We plot the variation of the adaptively selected $k$ values across training epochs. As shown in Fig. 9, $k$ starts relatively small, enabling the inclusion of more samples and thus stronger self-supervision signals at early stages. As training progresses, $k$ gradually increases, reflecting stricter neighborhood consistency criteria and leading the model to focus on more reliable high-confidence samples. This adaptive adjustment verifies the effectiveness of our method in balancing supervision quantity and quality throughout training.

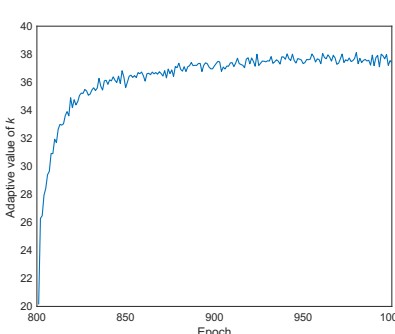

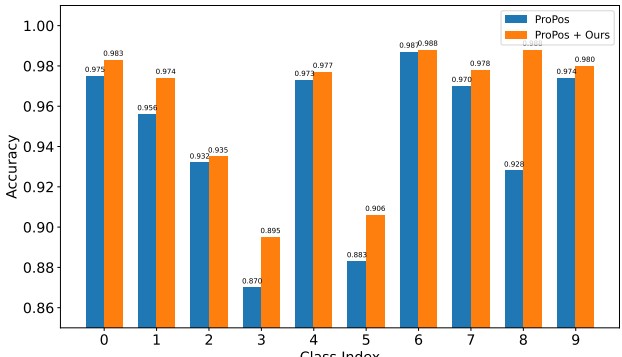

Figure 9: Visualizations of adaptive $k$.    Figure 10: Accuracy change for each class.

**Visualizations of accuracy change for each class.** To further analyze the effect of our method at the class level, we visualize the accuracy change of each category before and after applying DCBoost on CIFAR-10. The corresponding statistics are shown in Fig. 10. Our method consistently improves the accuracy across all 10 classes. These results demonstrate that our selection strategy does not overfit or bias toward specific categories, but instead achieves consistent gains across diverse semantic groups. Moreover, the improvements suggest that our method refines the global feature space in a class-balanced manner, ensuring that minority or challenging categories also benefit from the enhanced representations.

**Visualizations of high-confidence samples.** We visualize the feature space with t-SNE in Fig. 11, showing three subfigures: true labels, pseudo-labels, and high-confidence samples (red) versus others (blue). We observe that in regions where pseudo-labels disagree with ground truth (circled in the second subfigure), very few high-confidence samples are selected, meaning our method avoids unreliable areas. In well-formed clusters, high-confidence samples are densely distributed, showing that our method can reliably capture consistent regions. These observations indicate that our selection strategy **filters out noisy regions and focuses on structurally reliable clusters**, which in turn helps refine the global feature structure and improve clustering quality.

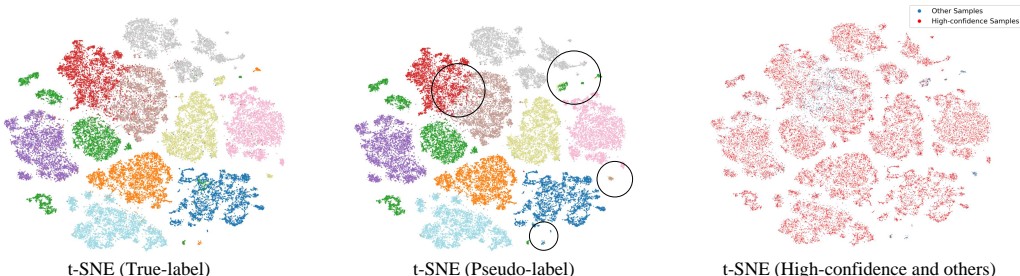

Figure 11: Visualizations of high-confidence samples on CIFAR-10.

