# OpenReview forum: "You Can Trust Your Clustering Model: A Parameter-free Self-Boosting Plug-in for Deep Clustering"
_NeurIPS.cc/2025/Conference — NeurIPS 2025 poster_

### Official Review · Reviewer_Tv3M · 2025-06-30

**Clarity:** 4
**Significance:** 2
**Originality:** 2
**Rating:** 5
**Confidence:** 3

**Summary:**

The paper proposes refining pre-trained Deep Clustering networks with plug-in losses, namely an instance consistency loss (inspired from self-supervisedlearning) and a positive (negative) loss that bring closer (futher) the representations of samples (or the class protypes of samples) based on their pseudo-labels. A central point in the paper is that the two latter losses are applied only on "reliable" samples -- those are identified as such based on consistency in the pseudolabels) of their k-NN neighboors. The method shows improvements in relation to the SOTA.

**Questions:**

I would appreciate to bring this work in the context of other works that perform sample selection/filtering in the domain of classification and in particular with the ones that utilize k-NN for doing so, and address the comment on scaling with the number of clusters.

"By incorporating an adaptive k-NN sample selection, our method strikes an optimal trade-off between the quality and quantity of self-supervision signals." It is not supported that the tradeoff is optimal in some sense. We experimentally see that it leads to improvemnts in some metrics, but that is not the same as claiming optimality.

**Ethical Concerns:**

["NO or VERY MINOR ethics concerns only"]

**Final Justification:**

The paper has novelty and good experimental results. The response addressed my concerns wrt to missing sota and experiments and I am happy to recommend acceptance.

**Limitations:**

yes

**Quality:**

3

**Strengths And Weaknesses:**

Positive:
-- The method is simple and the paper well written.
-- There is some novelty in the application of reliable sample sample selection and its use in the selective application of the losses in the domain of clustering.
-- The losses can be applied so as to refine existing models. Experiments show that when the losses are applied in several models, the improvements are consistent, albeit not always large.

Negative:

-- The paper essentially brings sample selection methods from the domain of classification to the domain of clustering. This has some novelty as I methioned in my view. However,  sample selection based on consistency of the labels of the samples in the local neighborhood is not new (e.g, [a1][a2]) -- I believe that the paper should at least refer to such works. Also, it is not clear whether other methods for sample selection, including the ones mentioned in the paper itself would give competitive performance. As it stands, I feel that the works in the domain learning with noisy labels (for classification) is not well covered.

-- The results presented are on datasets with a small number of clusters (10-20). It is unclear how the method scales with the number of clusters.

-- The method seems to need to perform k-means clustering. This introduces additional computational complexity.

[a1] Diego Ortego, Eric Arazo, Paul Albert, Noel E O’Connor, and Kevin McGuinness. Multi-objective interpolation training for robustness to label noise. In Proceedings of
the IEEE/CVF Conference on Computer Vision and Pattern Recognition, CVPR 2021.
[a2] Ssr: An efficient and robust framework for learning with unknown label noise, C Feng, G Tzimiropoulos, I Patras, BMVC, 2021

---

> ### Author Rebuttal · Authors · 2025-07-31
>
> We sincerely thank the reviewer for their constructive comments and thoughtful suggestions. We appreciate the recognition of the clarity and simplicity of our method. Regarding the main concerns, we provide detailed responses as follows:
>
> **N1&Q1-1. Relation to existing works in noisy label learning with local consistency-based selection.**
>
> Thanks for the reviewer's comments. **Sample selection is a fundamental and important problem**, especially in unsupervised learning and deep clustering. Related work can not diminish the novelty of our method, which **introduces clear innovations and provides advantages** beyond existing methods designed for the domain of noisy label learning.
>
> MOIT[a1] detects noisy labels by measuring the disagreement between a sample’s original label and the class distribution of its k-nearest neighbors in the global feature space. And SSR[a2] selects clean samples based on label–neighborhood consistency and uses a separate classifier to relabel samples with high confidence. **They all need do $k$-NN search in the whole dataset with a selected fixed k, and may have some hyperparameters like consistency and confidence thresholds.** However, our method is a parameter-free sample selection method and applies local adaptive k-nn within the mini-batch instead of the whole dataset.
> To provide a fair comparison, we **re-implemented MOIT and SSR** sample selection strategies and applied them on top of ProPos, replacing our selection module while keeping the rest of the training unchanged. Notably, SSR combines both $k$-NN filtering and confidence thresholding, which may not generalize to all baseline methods (e.g., those without a classifier head). Therefore, we only adopted its **$k$-NN-based filtering component** for compatibility.
>
>
> | Dataset         |          | CIFAR10  |          |          | CIFAR20  |          |          | ImageNet-Dogs |          |
> | --------------- | -------- | -------- | -------- | -------- | -------- | -------- | -------- | ------------- | -------- |
> | Method  | NMI | ACC  | ARI  | NMI  | ACC  | ARI  | NMI | ACC    | ARI |
> | ProPos          | 81.1     | 94.4     | 88.3     | 60.7     | 61.6     | 44.4     | 72.7     | 76.9          | 66.4     |
> | ProPos + Ours(MOIT selection) | 90.7     | 95.8     | 91.1     | 64.1     | 63.2     | 47.8     | 73.5     | 77.1          | 66.6     |
> | ProPos + Ours(SSR selection)    | 90.9     | 95.9     | 91.3     | 63.9     | 63.6     | 48.3     | 75.3     | 79.0          | 70.1     |
> | ProPos + Ours   | **91.1** | **96.0** | **91.6** | **64.5** | **63.9** | **49.2** | **76.3** | **79.7**     | **70.7** |
>
> To further ensure fairness with SSR, we conducted **additional CIFAR10 and ImageNet-Dogs experiments on the CDC baseline** by replacing our sample selection with full SSR filtering with confidence thresholding:
>
> | Dataset    |          | CIFAR10  |          |          | ImageNet-Dogs |          |
> | ---------- | -------- | -------- | -------- | -------- | ------------- | -------- |
> | Method |NMI |ACC|ARI|NMI |ACC|ARI|
> | CDC        | 89.3     | 94.9     | 89.5     | 76.8     | 79.2          | 70.2     |
> | CDC + Ours(SSR selection)  | 89.8     | 95.0     | 89.9     | 77.3     | 79.6          | 70.9     |
> | CDC + Ours | **89.9** | **95.1** | **90.0** | **77.5** | **79.7**   | **71.4** |
>
> Moreover, we compared their full pipelines with ours (including both filtering and loss function) on **CIFAR10** and observed that **the overall performance gain from MOIT and SSR remains consistently lower than ours,  even not better than ProPos**. Additionally, **our method achieves better results with shorter training time**, as shown below:
>
>
> | Method      | NMI      | ACC      | ARI      | Time(hour) |
> | ----------- | -------- | -------- | -------- | ---------- |
> | ProPos      | 88.1     | 94.4     | 88.3     | -          |
> | ProPos+MOIT[a1]       | 88.0     | 94.4     | 88.3     | -          |
> | ProPos+SSR[a2]         | 86.8     | 92.5     | 85.2     | -          |
> | ProPos+Ours(MOIT selection) | 90.7     | 95.8     | 91.1     | 5.0          |
> | ProPos+Ours(SSR selection)  | 90.9     | 95.9     | 91.3     | 4.8        |
> | ProPos+Ours | **91.1** | **96.0** | **91.6** | **4.5**    |
>
> While both MOIT and SSR improve over the baseline, their performance is **consistently inferior to our method**. We believe this is because **our framework is jointly optimized with the selection strategy and a discriminative loss**, which is not the case for methods like MOIT or SSR. When their selection is applied using their own loss to train the model, performance does not improve. In addition, **global $k$-NN filtering**, as used in MOIT and SSR, introduces extra computational cost, while our method remains **efficient and scalable** without additional complexity or hyperparameters.
> Following your suggestion, we will refer to the related works and add the results and analysis in the discussion part of our final paper.
>
>
>
> **N2&Q1-2. How the method scales with the number of clusters?**
>
> Thank you for the valuable comment on scalability. As shown in the main paper (Table 2), our method consistently outperforms baselines on datasets with 10–20 categories, which already demonstrates its strong performance in small- to medium-scale clustering scenarios.
>
> To further demonstrate the scalability of our method to datasets with a large number of categories, we additionally report results on **CIFAR100**, **Tiny-ImageNet (200 classes, more results are in Appendix B)**, and **ImageNet-1K (1000 classes)**. As shown below, our method consistently improves over the ProPos baseline across all datasets:
> | Dataset     |         | CIFAR100 |          |          | Tiny-ImageNet |          |          | ImageNet-1K |          |
> | ----------- | ------- | -------- | -------- | -------- | ------------- | -------- | -------- | ----------- | -------- |
> | Method | NMI | ACC  | ARI  | NMI  | ACC       | ARI  | NMI  | ACC     | ARI  |
> | ProPos      | 65.7    | 55.2     | 40.1     | 46.5     | 29.8          | 18.2     | 51.3     | 22.0        | 13.4     |
> | ProPos+Ours | **67.4** | **57.1** | **43.4** | **48.9** | **31.9**      | **19.8** | **54.5** | **23.2**    | **15.1** |
>
> These results confirm that our method is scalable and remains effective even when the number of clusters increases from tens to hundreds or thousands.
> **Importantly, our method is theoretically agnostic to the number of classes**. What fundamentally limits performance is not the number of categories itself, but the memory constraints that restrict batch size. When batch size is small relative to the number of classes, the chance of finding same-class neighbors per sample decreases. By introducing memory bank, our method compensates for this limitation without relying on strong alteration or architectural changes.
> And we will include these experiments in our final version.
>
>
> **N3. Additional computational complexity for k-means.**
>
> We appreciate the reviewer’s concern regarding the computational overhead introduced by $k$-means clustering. It is important to clarify that **$k$-means is not a contribution of our method but rather a commonly used approach for obtaining pseudo-labels** in representation-based deep clustering models. Moreover, **our method is flexible and does not rely strictly on $k$-means:** for clustering-head based models, we can replace $k$-means with a direct SoftMax classification layer to generate pseudo-labels.
>
> As demonstrated in Appendix C, Table 9, for the clustering-head baseline CDC, incorporating our method yields consistent improvements regardless of whether $k$-means or SoftMax is used for label assignment.
>
>
> **Q2. The tradeoff is not always optimal in some sense.**
>
> Thank you for pointing this out. We agree that the term "optimal" may overstate the result without theoretical or exhaustive empirical justification. We will revise the sentence in the paper to avoid this claim and now state that our adaptive $k$-NN sample selection achieves a _favorable_ trade-off between the quality and quantity of self-supervision signals, as demonstrated by consistent performance improvements in our experiments.

---

> > ### Author Response · Authors · 2025-08-04
> >
> > Dear Reviewer **Tv3M**,
> >
> > Thanks again for your time and effort in reviewing this paper. We appreciate your valuable comments to improve the quality of this work. If you still have concerns about this work, we would be more than happy to clarify them. Thanks.
> >
> > Regards from the authors.

---

> > > ### Author Response · Authors · 2025-08-06
> > >
> > > Dear Reviewer **Tv3M**,
> > >
> > > Thanks again for your time and effort in reviewing this paper. We appreciate your valuable comments to improve the quality of this work. **As the author-reviewer discussion period deadline is approaching**, please take five to ten minutes to check whether your concerns have been addressed.
> > >
> > > Thanks.

---

> ### Comment · Reviewer_Tv3M · 2025-08-06
> **Response**
>
> Thank you for the clarifications and the additional results. I am happy with the response. I can see that the method shows consistent improvement in relation to the ones I referred to and therefore i am raising my score

---

> > ### Author Response · Authors · 2025-08-06
> >
> > Dear Reviewer **Tv3M**,
> >
> > We are glad that your concerns are fully addressed. **Thanks for raising your score.** We will follow your suggestions to improve this paper in the final version. Thanks.
> >
> > Regards.

---

### Official Review · Reviewer_ZuXL · 2025-07-02

**Clarity:** 3
**Significance:** 2
**Originality:** 3
**Rating:** 4
**Confidence:** 4

**Summary:**

The paper proposes a plug-and-play design for improving a pre-trained clustering method iteratively by its confident predictions. This is achieved by an adaptive k-nn selection mechanism for identifying confident samples with consistent prediction as their neighbours. By explicitly pulling those confident and potentially positive sample pairs together in the latent space and pushing the centres of different clusters further away, the proposed method is shown effective on boosting the clustering performance of a wide range of pretrained models.

**Questions:**

Since all the additional constraints introduced to the pre-trained model are applied to the features instead of their clustering predictions, it is interesting to see how the improvements made in this paper help with other representation learning benchmarks, e.g., are the improved features more generalisable to new data domains or new tasks?

Please also refer to the strengths and weaknesses section, where my main concerns are listed.

**Ethical Concerns:**

["NO or VERY MINOR ethics concerns only"]

**Final Justification:**

The discussion about other self-labelling methods and comparison to them is convincing, which addresses my primary concerns about the effectiveness of the proposed model. The additional runtime information provided by the authors answers my questions about efficiency. The evaluation of the feature's generalisation ability to other tasks demonstrates the potential impact of this work beyond clustering. Given all these, I am happy to improve my rating of this work.

Besides, I would still like to encourage the authors to dig deeper into the performance degradation issue when applying other self-labelling methods to pre-learned features, e.g., whether this can be avoided by freezing the network partially or at the early training stage.

**Limitations:**

Yes

**Quality:**

2

**Strengths And Weaknesses:**

Strengths:
+ The paper is generally well-written with clear motivation for improving the global cluster structure by the potentially reliable pairwise relationships drawn from the local structure.
+ The proposed method is effective and has been shown generalisable to benefit extensive pre-trained models on various benchmarks.

Weaknesses:
+ Important related works & baselines missing. The proposed method follows the self-training/self-labelling paradigm to gradually improve itself with its own predictions. Such an idea is commonly seen in unsupervised training and deep clustering [1][2]. Those types of self-training/self-labelling methods can be applied to pre-trained models to obtain further improvements. There is only CoNR being discussed from this perspective, but a more comprehensive study and comparisons is essential.
+ The “adaptive” k-NN filtering sweeps over all candidate values of “k” can be inefficient, especially when dealing with large-scale training data. The performance improvements it brings shall be further studied together with the increase in training cost.
+ How does the positive loss help with mitigating imbalanced clustering is unclear to me. How does it help with assigning each class an equal contribution as claimed at L248?
+ It is also unclear how the cluster prototypes are computed. What will happen to the clusters that have no sample in a mini-batch? This can happen when the cluster distribution is imbalanced or the number of clusters is greater than the batch size.


[1] Caron, Mathilde, et al. "Deep clustering for unsupervised learning of visual features." Proceedings of the European conference on computer vision (ECCV). 2018.
[2] Gansbeke, Van, et al. “Scan: Learning to classify images without labels.” Proceedings of the European Conference on Computer Vision (ECCV). 2020

---

> ### Author Rebuttal · Authors · 2025-07-31
>
> We sincerely thank the reviewer for the careful evaluation and constructive feedback on our work. Below, we provide detailed responses to each of the key concerns raised.
>
> **W1. More self-labelling paradigm.**
>
> Thanks for your insightful comment. Our paper has already included comparisons with CoNR, a more recent and relevant representative of this paradigm. Follow the reviewer's suggestion, we additionally implemented **self-labeling pipelines based on DeepCluster[1] and SCAN[2]** on the ProPos baseline. Specifically, for SCAN, we used pretrained ProPos representations and performed the second and third training phases of SCAN; for DeepCluster, we attached a randomly initialized clustering head and supervised the model using self-generated pseudo-labels.
>
> The results on **CIFAR10** (shown below) indicate that **these self-labeling methods do not improve performance** over the ProPos baseline. In contrast, our method consistently yields significant gains. This suggests that existing self-labeling paradigms may not always transfer effectively to pretrained models.
>
> |Method|NMI|ACC|ARI|
> |-|-|-|-|
> |ProPos|88.1|94.4|88.3|
> |ProPos+SCAN-2[2]| 87.3|93.0|85.9|
> |ProPos+SCAN-2+SCAN-3[2]|87.1|93.1|86.0|
> |ProPos+DeepCluster[1]|86.5|93.6|86.6|
> |ProPos+Ours|**91.1**|**96.0**|**91.6**|
>
> One potential reason why DeepCluster and SCAN underperform is **their reliance on joint training with a randomly initialized clustering head**. Prior work CDC[3] has shown that **attaching randomly initialized heads can destabilize pre-trained representations**. Therefore, while these methods work well in end-to-end pipelines with consistent optimization objectives, **they may degrade performance when grafted onto pretrained models without adaptation**. By contrast, our method preserves the pretrained model and only introduces a lightweight asymmetric predictor, which minimizes disruption and maintains training stability. This design enables effective improvement on strong pretrained features without architectural modification.
>
> We hope this clarifies that off-the-shelf application of clustering-head based self-labeling methods is not universally effective, and underscores the novelty and necessity of our method. Full results and analysis will be included in the discussion part of main paper in the final version.
>
> **W2. About training cost of adaptive k-NN filtering.**
>
> Our adaptive $k$-NN filtering introduces **negligible computational overhead**. Specifically, the top‑$m$ neighbors are retrieved **only once per batch**, and scores for all candidate $k$ (from 1 to $m$) are computed **based on this single result**. The optimal-$k$ selection relies on **simple, vectorized label consistency checks over the retrieved indices** (see function $find optimal-k-balanced$  in code). This process has a time complexity of **$O(B·m)$**, where $B$ is the batch size and $m$ is the maximum number of neighbors considered. Notably, **this step is independent of the total dataset size $N$**. Therefore, our method avoids the inefficiencies typically associated with exhaustive sweeping over multiple k values and remains scalable to large datasets.
>
> To validate this, we have conducted experiments on CIFAR-10 with settings: using no filtering ($k$ = 0), fixed $k=25$, and adaptive selection with $m=25,50,75$. As shown below, **training time remains nearly identical across all variants**, but get great performance improvements:
>
> |Variant |NMI|ACC|ARI|Time(h:min:s)|
> |-|-|-|-|-|
> |ProPos|88.1|94.4|88.3|-|
> |$k=0$|89.1|94.9|89.3|4:22:32|
> |$k=25$|91.1|96.0|91.6|4:25:58|
> |$m=25$|91.2|96.1|91.7|4:25:51|
> |$m=50$|91.1|96.0|91.6|4:29:00|
> |$m=75$|91.1|96.0|91.6|4:25:58|
>
> The results demonstrate that our adaptive filtering strategy is both **efficient and effective**, providing performance benefits without introducing additional training cost.
>
> **W3. How does the positive loss help with mitigating imbalanced clustering?**
>
> The **positive loss** is designed to promote intra-class compactness among **high-confidence samples**. However, some classes may dominate this loss simply because they have more confident samples. To address this, we introduce a **per-class normalization weight** $w_c$, which ensures that **each class contributes equally** to the overall loss, regardless of its size. Specifically, $w_c$ can be seen as the **inverse** of the square of the L2 norms of the aggregated features from class c. **When a class has more high-confidence samples (thus a larger summed feature norm), its influence on the loss is automatically reduced.** In this way, the loss naturally compensates for the **imbalance in the number of high-confidence samples** across classes.
>
> **W4-1. How to compute cluster prototypes?**
>
> As described in the paper, cluster prototypes are computed by **averaging the features of high-confidence samples within each pseudo-label class in current training batch**, followed by L2 normalization. The number of prototypes $c_B$ in a batch corresponds to the number of classes with confident samples present in that batch.
>
> **W4-2. What will happen to the clusters that have no sample in a mini-batch?**
>
> If a cluster has no sample in a mini-batch **when the class distribution is imbalanced,** we simply exclude that class from the current loss computation. This design does not affect the stability of training, since the model learns from the cumulative gradients over many batches rather than requiring every batch to contain all classes.
>
> Importantly, we additionally validated the method's robustness on long-tailed benchmarks CIFAR10-LT and CIFAR20-LT, where class sizes follow an **exponential decay** (e.g., with ratio=10, head has 5000 samples, tail only 500).Our method still consistently outperforms the baselines. This demonstrates that the occasional absence of some classes in individual batches does not lead to significant degradation in overall clustering performance.
>
> |Ratio=10||CIFAR10-LT|||CIFAR20-LT||
> |-|-|-|-|-|-|-|
> |Method| NMI| ACC|ARI|NMI|ACC|ARI|
> |ProPos|57.1|48.3|40.7|47.6|42.1|29.9|
> |ProPos+Ours|**61.3**|**49.9**|**43.3**|**49.1**|**43.4**|**31.5**|
> |CoNR|67.2|64.6|56.7|51.2|43.9|30.3|
> |CoNR+Ours|**68.4**|**65.3**|**59.1**|**51.4**|**44.1**|**33.3**|
> |LFSS [4]|57.9|56.2|43.0|46.7|41.4|28.3|
> |LFSS+Ours|**61.3**|**60.1**|**46.9**|**47.5**|**42.4**|**29.4**|
>
> We added per-class accuracy analysis on **CIFAR20-LT**,  where C0 is the most frequent (head) class and C19 is the rarest (tail) class. We compared the class-wise accuracy before and after integrating our method:
> |Method\Class|C0|C1|C2|C3|C4|C5|C6|C7|C8|C9|C10|C11|C12|C13|C14|C15|C16|C17|C18|C19|
> |-|-|-|-|-|-|-|-|-|-|-|-|-|-|-|-|-|-|-|-|-|
> |ProPos|33.1|24.5|58.6|30.9|66.9|25.1|43.3|69.0|37.3|62.2|73.9|53.1|33.6|6.8|81.2|12.4|1.4|90.9|0.0|0.8|
> |ProPos+Ours|**34.9**|**26.3**|**59.8**|**31.5**|**69.5**|23.0|43.1|**70.4**|36.5| **66.4**|**74.2**| 52.5| **36.2**|**7.5**| **84.7**|**14.6**|1.4|**91.6**|**12.4**|0.0|
>
> **Our method brings improvements in the majority of classes; importantly, it is not restricted to head classes only**. Several tail classes, which are more likely to be absent from some mini-batches, still get noticeable performance gains.
> Beyond class imbalance, we acknowledge the challenges of scaling to many clusters. Our primary focus is on moderate-scale settings where batch sizes can cover most classes. In response to the reviewer’s concern, we have newly added experiments in a setting where **the number of clusters exceeds the batch size**. Although our base selection method may face limitations due to batch size constraints, it can still be extended to these challenging settings by incorporating a memory bank to store and reuse features across batches, and adopting a fixed $k$-NN retrieval method to stabilize neighbor selection.
> We add experiments on **CIFAR100**, where we intentionally reduce the batch size to 64 (i.e., fewer than the number of classes) to simulate this, and size 256 on **ImageNet-1K**, a large-scale dataset with 1,000 categories. Because full training of ImageNet-1K is resource-intensive, we used ProPos to train for 30 epochs and added our method for another 10 epochs.
> |Dataset||CIFAR100|||ImageNet-1K||
> |-|-|-|-|-|-|-|
> |Method|NMI|ACC|ARI|NMI|ACC|ARI|
> |Propos|65.7|55.2|40.1|51.3|22.0|13.4|
> |Propos+Ours|**67.1**|**57.4**|**43.8**|**54.5**|**23.2**|**15.1**|
>
> Our method still improves clustering performance across all metrics on both datasets. It is also indicated that **using high-quality local information to guide the optimization of global structure** remains effective, even when not all classes are present in each mini-batch. We will include the results and analysis in the final version of our paper.
>
> **Q1. Are the improved features more generalizable to new data domains or new tasks?**
>
> To evaluate whether our improved representations are more generalizable, we follow standard linear evaluation protocols. We freeze the learned features and train a linear classifier on top of them for downstream classification. Specifically, we conduct two experiments:
>
> (1) **In-domain generalization:** Linear classification on CIFAR10 using features learned on training set of CIFAR10.
> (2) **Cross-domain generalization:** Linear classification on STL10 using features learned on CIFAR10.
>
> We report Top-1 accuracy and Top-5 accuracy as performance metrics.
>
> As shown below, our method consistently improves performance over the baseline in both settings, suggesting that the enhanced features are not only more discriminative within the same domain, but also more transferable to related domains.
>
> |CIFAR10|Top1|Top5|
> |-|-|-|
> |ProPos|94.7|99.8|
> |ProPos+Ours|**95.0**|99.8|
>
> |STL10|Top1|Top5|
> |-|-|-|
> |ProPos|76.2|98.8|
> |ProPos+Ours|**77.6**|98.6|
> |CoNR|67.0|96.2|
> |CoNR+Ours|**73.6**|**98.1**|
>
> [3]Towards Calibrated Deep Clustering Network, ICLR, 2025.
>
> [4]Learning from Sample Stability for Deep Clustering, ICML, 2025.

---

> > ### Author Response · Authors · 2025-08-04
> >
> > Dear Reviewer **ZuXL**,
> >
> > Thanks again for your time and effort in reviewing this paper. We appreciate your valuable comments to improve the quality of this work. If you still have concerns about this work, we would be more than happy to clarify them. Thanks.
> >
> > Regards from the authors.

---

> > > ### Author Response · Authors · 2025-08-06
> > >
> > > Dear Reviewer **ZuXL**,
> > >
> > > Thanks again for your time and effort in reviewing this paper. We appreciate your valuable comments to improve the quality of this work.  **As the author-reviewer discussion period deadline is approaching**, please take five to ten minutes to check whether your concerns have been addressed.
> > >
> > > Thanks.

---

> > ### Comment · Reviewer_ZuXL · 2025-08-06
> >
> > Thanks for the detailed explanation. The authors' clarification addressed most of my concerns about model designs and efficiency.
> >
> > In terms of the missing related works, it is surprising to see that applying SCAN's self-labelling makes the performance of the baseline worse. To my understanding, SCAN's clustering is also built on top of pre-learned visual representations. Besides, I would suggest adding not only quantitative comparisons but also a more thorough discussion about deep clustering with self-labelling in the manuscript. This is essential to distinguish the proposed method from existing works regarding how the pseudo-labels are constructed and how the unreliable supervision signals are avoided.
> >
> > It is appreciated that additional efforts are made to verify the generalisation of feature representations to other classification tasks. In the future version, I believe extending the scope to even more types of tasks, like detection and segmentation, will make this work stronger and bring broader impacts.

---

> > > ### Author Response · Authors · 2025-08-06
> > >
> > > Thank you for your insightful comments and suggestions.
> > >
> > > Regarding the concern that SCAN is also built on top of pre-trained visual representations, we would like to clarify the key difference. Although SCAN starts from pre-trained representations, its second and third stages involve **joint training with a randomly initialized clustering head**, and the cluster predictions are obtained through this head. The design of **randomly initializing the clustering head** is **unstable and will destroy the feature learned by the pre-trained model**: in our experiments, after adding the clustering head, the performance initially **drops from ~94% to ~87%**. **Although it improves during the training stages, it only recovers to around 93%**, still lower than the baseline. This suggests that the performance gain from SCAN's self-labeling process **cannot fully offset the degradation introduced by the randomly initialized head.** In contrast, **our method does not require any clustering head**—cluster assignments are obtained directly from the **representations of the target network via k-means**, preserving the stability of the pre-trained features. Furthermore, we enhance the training process **by selecting high-confidence samples based on reliable local information, and applying a discriminative loss on these samples.** This selective self-supervision further stabilizes training and improves clustering performance.
> > >
> > > Following your suggestion, we will revise the manuscript to include a more detailed discussion on self-labeling-based deep clustering methods in the final version. We will clarify how our method differs in terms of **pseudo-label construction and training stability,** especially in comparison to SCAN and similar pipelines.
> > >
> > > We also appreciate your suggestion to extend this line of work to broader tasks such as detection and segmentation, and we will mention this as a valuable direction in the discussion section.

---

> > > ### Author Response · Authors · 2025-08-07
> > >
> > > Dear Reviewer **ZuXL**,
> > >
> > > Thanks again for your valuable comments. We are glad that most of your concerns are addressed. We will follow your suggestions to extend the scope of our method to more types of tasks. In the previous response, we explained why applying SCAN's self-labelling directly may sometimes make the performance of the baseline worse. Hope the explanations will fully address your concerns. If you still have concerns, please let us know, and we will try our best to clarify. If you did not have other concerns, please consider raising your final rating. Thanks a lot.
> > >
> > > Regards from the authors.

---

### Official Review · Reviewer_S4wz · 2025-07-03

**Clarity:** 2
**Significance:** 2
**Originality:** 3
**Rating:** 4
**Confidence:** 3

**Summary:**

This paper presents DCBoost, a parameter-free plug-in module designed to enhance the global structure of deep clustering models by leveraging local neighborhood consistency. I believe using trustworthy local cues to guide global optimization is intuitive, novel, and practically relevant. The authors demonstrate improvements across several popular baselines and datasets. However, the work still exhibits several limitations in terms of robustness under real-world conditions, such as noise and imbalance, limited generalization beyond flat clustering, and incremental performance gains despite extensive evaluation.

**Questions:**

- Your method assumes local k-NN consistency is a reliable cue for selecting high-confidence samples. However, under class imbalance or label noise, the local neighborhood may become less trustworthy.
- Is DCBoost applicable to hierarchical or density-based clustering models? If not, what are the obstacles?
- Could you provide insights into real-world scenarios (e.g., medical, industrial) where such gains are critical?

**Ethical Concerns:**

["NO or VERY MINOR ethics concerns only"]

**Final Justification:**

The author's answer resolved some of my concerns. I think it is good work. However, clustering should not only focus on accuracy, but also explore the discoveries or generalizations brought about by applications in real scenarios. This is also why deep clustering methods have not been popular. In practice, we prefer traditional clustering methods such as k-means.

**Limitations:**

- The assumption that the number of clusters is known in advance.
- Potential performance degradation under imbalanced or non-uniform data distributions.
- The lack of mechanisms to dynamically estimate or adapt to unknown class counts.

**Quality:**

3

**Strengths And Weaknesses:**

**Strengths:**
- The work uses adaptive k-NN consistency to guide representation learning, which is a fresh and valuable perspective.
- The proposed module does not modify network structures, making it highly practical for real-world and legacy systems.
- The paper includes ablation studies on loss terms, adaptive k selection, and class weighting strategies, showing clear empirical support for each design choice.

**Weaknesses:**
- The method assumes that local structures are reliable, yet this assumption may break under class imbalance or noisy samples.
- The framework seems incompatible or untested on models based on hierarchical clustering or density-based clustering, limiting generality.

---

> ### Author Rebuttal · Authors · 2025-07-31
>
> Thank you for the reviewer’s constructive and thoughtful feedback. Below, we address each of your concerns one by one:
>
> **About incremental performance gains in deep clustering.**
>
> We would like to emphasize that **DCBoost achieves consistent and significant performance improvements across a wide range of strong baselines**, despite already high starting points. This indicates that our method is capable of **further enhancing state-of-the-art results**, which is non-trivial in the unsupervised clustering domain.Importantly, under **strictly unsupervised conditions**, our method establishes **new state-of-the-art performance** across multiple standard benchmarks without using any true labels. In this context, **the observed gains are both meaningful and substantial.**
>
>
> **W1&Q1. The local neighborhood may become less trustworthy under class imbalance or label noise.**
>
> Thank you for the insightful comment regarding the potential unreliability of local neighborhood consistency under class imbalance. But we would like to clarify that in our **unsupervised clustering** setting, the model does **not rely on ground-truth labels**, and thus is **not subject to conventional label noise** originating from human annotation errors. Instead, the labels are generated via self-supervised learning or clustering, which may contain **inherent uncertainty**, but this differs fundamentally from traditional noisy label scenarios. Therefore, while the neighborhood consistency may be affected by imperfect pseudo-labels, the concern of **external label noise** corrupting local structure is **not applicable** here.
>
> **Regarding class imbalance**, we have added a qualitative analysis of the local $k$-NN neighborhood structure on long-tailed dataset CIFAR10-LT and CIFAR20-LT, where the number of training samples per class follows an **exponential decay** controlled by a ratio between the number of samples in the most frequent class and the least frequent class. For instance, with ratio=10, the head class contains 5000 samples while the tail class has only 500.  As expected, local consistency does degrade slightly compared to the balanced case due to skewed class distributions. However, it still remains **significantly more reliable than global structural signals**, making it still a valuable cue for identifying high-confidence samples. Specifically, our results show that $k$-NN accuracy remain much higher compared to the current clustering ACC and better than global features, even in long-tailed scenarios:
>
> | CIFAR10-LT    | Intra-class ↑ | Inter-class ↓ | KNN Accuracy(%) ↑ | SC↑ | ACC (%) ↑ |
> | ------------- | ------------- | ------------- | -------------- | ------ | --------- |
> | ProPos | 0.188        | -0.017       | 87.5| 0.081  | 46.1      |
> | ProPos + Ours | **0.924**        | **-0.101**       | **95.4** | **0.682** | **47.4**|
>
> | CIFAR20-LT    | Intra-class ↑ | Inter-class ↓ | KNN Accuracy(%) ↑ | SC ↑ | ACC (%) ↑ |
> | ------------- | ------------- | ------------- | -------------- | ----- | --------- |
> | ProPos        | 0.156        | -0.009  | 70.3 | 0.088 | 42.1|
> | ProPos + Ours | **0.775**        | **-0.038**       | **70.8**  | **0.487** | **43.4** |
>
> To further validate the robustness of our method, we conducted additional experiments on **CIFAR10-LT and CIFAR20-LT**. The following results demonstrate that our method consistently improves clustering performance over strong baselines:
>
> | Dataset (Ratio=10) |          | CIFAR10-LT |          |          | CIFAR20-LT |          |
> | ------------------ | -------- | ---------- | -------- | -------- | ---------- | -------- |
> | Method             | NMI      | ACC        | ARI      | NMI      | ACC        | ARI      |
> | ProPos        | 57.1     | 48.3       | 40.7     | 47.6     | 42.1       | 29.9     |
> | ProPos+Ours    | **61.3** | **49.9**   | **43.3** | **49.1** | **43.4**   | **31.5** |
> | CoNR           | 67.2     | 64.6       | 56.7     | 51.2     | 43.9       | 30.3     |
> | CoNR+Ours      | **68.4** | **65.3**   | **59.1** | **51.4** | **44.1**   | **33.3** |
> | LFSS [1]           | 57.9     | 56.2       | 43.0     | 46.7     | 41.4       | 28.3     |
> | LFSS+Ours      | **61.3** | **60.1**   | **46.9** | **47.5** | **42.4**   | **29.4** |
>
> These results demonstrate that our $k$-NN-based selection strategy is **effective and robust, even when local consistency may be affected by class imbalance.**  Although we do not incorporate any specific mechanisms to address non-uniform data distributions scenarios, our method **still yields consistent improvements** across different datasets and baselines.
>
>
> **W2&Q2. Generality on the hierarchical clustering or density-based clustering model.**
>
> While our method was not originally tailored for hierarchical or density-based clustering—since flat clustering remains the mainstream in deep clustering—we recognize the importance of generalizability across paradigms. To this end, we have conducted additional experiments on CIFAR10 using a hierarchical clustering model. Although DCBoost does not leverage the hierarchical structure explicitly, we only retained the **pre-hierarchical representation part** of the model and integrated it into our own training framework. Based on these learned representations, we applied our training process. **Even without utilizing the full hierarchical pipeline**, DCBoost consistently improved the clustering performance, **outperforming both the original representation and the full hierarchical clustering baseline**:
> | Method                             |NMI      | ACC      | ARI      |
> | ---------------------------------- | -------- | -------- | -------- |
> | CoHiClust [2] (representation only)     | 75.5     | 82.5     | 70.5     |
> | CoHiClust(hierarchical clustering) | 77.8     | 83.7     | 72.9     |
> | CoHiClust(representation)+Ours | **77.9** | **84.4** | **73.6** |
>
> Extending **DCBoost** to **density-based clustering methods** (such as DBSCAN) presents additional challenges. These methods are **non-parametric** and do not assume a fixed number of clusters, making it difficult to align with DCBoost’s current reliance on pseudo-labels. However, adapting DCBoost to such methods remains a promising direction for future work.
>
>
> **Q3. Insights into real-world scenarios.**
>
> We thank the reviewer for highlighting the importance of real-world relevance. To this end, we evaluated our method on **PathMNIST**, a medical imaging dataset derived from colon pathology slides, involving the classification of different tissue types. Our method shows significant improvements over the baseline ProPos:
> | Method                             |NMI      | ACC      | ARI      |
> | ---------------------------------- | -------- | -------- | -------- |
> | ProPos|58.4| 67.6| 48.6|
> | ProPos+Ours | **63.7**|**72.9**| **55.3**|
> The consistent improvements across all metrics suggest our method achieves more accurate and robust clustering. Such gains can help provide more stable and consistent pseudo-labels in weakly supervised or semi-supervised clinical workflows, and enhance the reliability of AI-assisted diagnosis or pre-screening.
> Beyond this,  in **industrial anomaly detection**, where labeled anomalies are rare, selecting trustworthy normal patterns through local consistency improves unsupervised model robustness. These scenarios rely on strong pseudo-supervision, making our framework’s improvements especially valuable in practice.
>
>
> **Limitation in estimating class counts.**
>
> **We would like to point out that the vast majority of existing deep clustering methods assume the number of clusters K as a predefined input.** Similarly, our method also operates under this common assumption. However, we included over-clustering and under-clustering experiments in Appendix B, Table 8 (e.g., CIFAR-20 with cluster numbers set to 10, 20, 30, 40, 50) that our method remains robust even when the specified cluster number deviates from the true number of classes. In particular, DCBoost consistently outperforms the baseline ProPos across all settings. These results suggest that, although our method does not explicitly estimate the number of clusters, it remains more robust to such inaccuracies compared to baseline methods.
> We agree that incorporating mechanisms to dynamically estimate or adapt to unknown class counts is a valuable direction and consider this an interesting avenue for future work.
>
> [1] Learning from Sample Stability for Deep Clustering, ICML, 2025.
>
> [2] Contrastive Hierarchical Clustering, ECML-PKDD, 2023.

---

> > ### Author Response · Authors · 2025-08-04
> >
> > Dear Reviewer **S4wz**,
> >
> > Thanks again for your time and effort in reviewing this paper. We appreciate your valuable comments to improve the quality of this work, and your positive recommendation. If you still have concerns about this work, we would be more than happy to clarify them. Thanks.
> >
> > Regards from the authors.

---

> > ### Comment · Reviewer_S4wz · 2025-08-04
> >
> > The author's answer resolved some of my concerns. I think it is good work. However, clustering should not only focus on accuracy, but also explore the discoveries or generalizations brought about by applications in real scenarios. This is also why deep clustering methods have not been popular. In practice, we prefer traditional clustering methods such as k-means.

---

> > > ### Author Response · Authors · 2025-08-05
> > >
> > > Dear Reviewer **S4wz**,
> > >
> > > Thanks for your recognition of our work. We fully agree on the importance of exploring discoveries and generalizations brought about by clustering in real-world applications. However, regarding the concern that deep clustering has not been widely adopted compared to traditional methods such as k-means, we would like to clarify that **deep clustering is not a replacement for traditional clustering but an enhancement**: it **jointly learns the feature space and cluster assignments**, making conventional algorithms like k-means far more effective on complex data such as images.
> > >
> > > For example, directly applying **k-means on raw CIFAR-10 images yields only ~20% accuracy** due to the lack of meaningful structure in the pixel space. In contrast, applying **k-means to features learned via deep clustering achieves up to 96% accuracy (our method)**. This drastic improvement demonstrates the benefit of deep clustering. When clustering performance is poor, the learned features are unlikely to capture useful semantic structure—making meaningful discovery or generalization even harder. Deep clustering overcomes this by learning feature spaces where clusters are more separable and semantically meaningful, thus **laying a stronger foundation for unsupervised knowledge discovery**. It is also worth noting that **the final clustering step in deep clustering remains compatible with traditional algorithms like k-means.**
> > >
> > > Moreover, we respectfully note that deep clustering **does not conflict with the goal of discovery or generalization in real-world applications**—on the contrary, it facilitates it. By learning task-relevant features, **deep clustering uncovers more interpretable and actionable structures in the data**. For instance, our experiments on a medical imaging dataset (rebuttal to Q3) show that even without labels, **the learned clusters align well with meaningful patterns, supporting unsupervised knowledge discovery.**
> > >
> > > Regards from the authors.

---

### Official Review · Reviewer_EzRr · 2025-07-05

**Clarity:** 3
**Significance:** 2
**Originality:** 3
**Rating:** 4
**Confidence:** 4

**Summary:**

This paper presents a novel deep clustering enhancement method, DCBoost, aimed at improving clustering performance by refining the global structure of existing deep clustering models. DCBoost is a plug-in that does not require manual hyperparameter tuning, and it leverages local structure information to optimize the global feature structure, thereby enhancing clustering accuracy. Specifically, the method employs an adaptive k-nearest neighbor (k-NN) approach to select high-confidence samples, and discriminative loss is computed based on these samples to enhance intra-class compactness and inter-class separability during training. Experimental results show that DCBoost significantly improves the performance of existing deep clustering models across multiple benchmark datasets and demonstrates strong generalization ability across various datasets.

**Questions:**

1.	The description of the dynamic selection mechanism is unclear. It is not specified whether retrieval is performed only on the target network’s output or if both target and online network outputs are considered. More implementation details are needed;
2.	Is dynamic retrieval necessary for every batch? If so, does it require saving historical sample representations? The method for updating these representations should also be discussed;
3.	The choice of setting the dynamic retrieval parameter m to 50 needs further justification. Does this lead to excessive computational complexity? A comparison of time consumption, especially with a fixed m, would clarify the computational trade-offs;
4.	How can the method ensure the selection of high-confidence samples from different classes in each batch? It would be helpful if the authors could show how clustering precision changes across different classes before and after applying the method;
5.	Regarding the score visualization, the authors explain the score as the area of the shaded rectangle, which is intuitive. However, replacing the x-axis with precision could better illustrate the trade-off between quality and quantity, which warrants further exploration;
6.	How does the proposed method compare with existing high-quality sample selection methods in terms of accuracy and computational efficiency? What is the time complexity of the proposed method compared to others?

**Ethical Concerns:**

["NO or VERY MINOR ethics concerns only"]

**Final Justification:**

My primary concerns have been resolved, and I will update my rating accordingly.

**Limitations:**

The authors have acknowledged the following limitations in the main text.
“A potential limitation is that our method assumes the number of clusters is known and lacks mechanisms to handle highly imbalanced or non-uniform data distributions, which may reduce its adaptability in complex real-world scenarios.”

**Quality:**

2

**Strengths And Weaknesses:**

Strengths

1.	The paper proposed a novel adaptive k-NN method for selecting high-confidence samples, effectively addressing the challenge of choosing the optimal k in clustering tasks. It aims to strike an optimal balance between the quality and quantity of self-supervision signals, thus improving clustering performance.
2.	The paper provides extensive comparison and ablation studies, thoroughly validating the effectiveness of the proposed method in improving the performance of existing deep clustering models.

Weaknesses

1.	Certain parts of the paper, such as the Adaptive k-NN Filtering in Section 3.1, require further clarification to enhance understanding of the method's detailed operation.
2.	The experimental setup, including batch size and total training epochs, is not sufficiently detailed. This information is essential for understanding the reproducibility and scalability of the experiments.
3.	Some method implementations, such as those referenced in Table 2 (e.g., CC, ProPos), are unclear. It is not specified whether the proposed method modifies the backbone network or if it requires additional components like a projection layer or clustering head.

---

> ### Author Rebuttal · Authors · 2025-07-31
>
> We sincerely thank the reviewer for their valuable comments and constructive feedback. We would like to address each of your concerns in detail, and clarify points that may have caused confusion.
>
> **W1&Q1. The description of the dynamic selection mechanism.**
>
> Thanks the reviewer for pointing out the need for more clarity regarding our dynamic selection mechanism. We clarify the implementation details as follows:
>
> **For BYOL-like architecture (BYOL, CoNR, ProPos):** We combine outputs from both **the online and target networks.**  The similarity matrix $S$ is computed as: $S_{ij}=\cos\left(z^o_i, z^t_j\right)$ to search the $k$-NN. This hybrid approach balances the stability of the target network, and the real-time adaptability of the online network. By using both, the neighborhood consistency reflects both stable semantic structure and recent representation shifts.
>
> **For other architectures(e.g., CC, SCAN, CDC):** We only use the **target network's features** for $k$-NN retrieval. ($S_{ij}=\cos\left(z^t_i, z^t_j\right)$). This is because the predictor followed by the online network is randomly initialized, and may introduce noise or unstable representations early in training. To ensure reliable consistency estimation, we rely solely on the more stable target encoder output in such cases.
> We will revise the manuscript accordingly to clearly describe these design choices and implementation details.
>
>
> **W2. Experimental setup about batch size and total training epochs.**
>
> In fact, we have already provided comprehensive information regarding the experimental setup in **Appendix A**, including backbone architectures, dataset sizes, training durations, batch sizes, and other relevant hyperparameters. For clarity, we would like to reiterate here that our method uses a **batch size of 256** and is trained for **200 epochs** based on the existing models. We believe this level of detail is sufficient to ensure the **reproducibility and scalability** of our experiments. **Besides, we have submitted our code in the supplementary file**, which makes the reproducibility easy.
>
>
> **W3. Whether the proposed method modifies existing models.**
>
> Our proposed method is applied directly on the existing models without modifying their backbone networks or any other components. **As detailed in Appendix A**, for models already based on BYOL-like architecture such as BYOL, CoNR, and ProPos, **neither the backbone nor additional modules are changed or added** during our enhancement. For other models like CC,SCAN and CDC, we duplicate their clustering network to form online and target networks, and add only **a lightweight MLP predictor after the online network**. Still, the backbone networks and original modules remain unaltered.
>
>
> **Q2. Is dynamic retrieval necessary for every batch?**
>
> Yes, our method performs  $k$-NN-based clean sample selection at **each training batch**. That means, **if the batch size is $B$, we compute $k$-NN using only $B$ samples.** By default, we perform **within-batch retrieval**, which is efficient and sufficient for all the results reported in the main paper. Thus, **no historical representations are used in these main experiments.**
>
> Instead, to handle **rare edge cases** (e.g., when a batch lacks sufficient same-class samples), we introduce a **memory bank** of recent features from the **target network**. This bank is continuously updated with the most recent representations at each training step. This mechanism is only used in the extended experiment in Appendix B, and does not involve maintaining long-term historical representations.
>
>
> **Q3. The dynamic retrieval parameter m setting.**
>
> We clarify that evaluating multiple candidate values of $k$ (from 1 to $m$) **does not require performing $k$-NN retrieval multiple times**. Instead, **we compute the $m$-nearest neighbors only once** per batch (e.g., $m=50$), and **all scores under different $k$ are evaluated based on this single retrieval result.** The function $find optimal-k-balanced$ (see code) performs label consistency checks and computes an favorable $k$ using efficient matrix operations. Therefore, if the number of samples within the batch is $B$, the overall computational complexity of dynamic retrieval remains $O(B·m)$ per batch, so the whole computational complexity of training is not significantly affected by the value of $m$.
>
> We have also added a new experiment on CIFAR10 varied $m$ (e.g., $m$=25, 50, 75), replaced the dynamic strategy with a fixed $k$=25, and even removed the $k$-NN-based filtering altogether (i.e., $k=0$, using all samples) as follows. **All these variants showed negligible difference in runtime.** This confirms that our strategy introduces **no observable computational overhead**, while effectively improving sample selection.
> |Variant|NMI|ACC|ARI|Training Time (h:m:s)|
> |--|--|--|--|--|
> | ProPos|88.1|94.4|88.3|-|
> |k = 0| 89.1|94.9| 89.3|4:22:32|
> |k = 25|91.1|91.6 |96.0|4:25:58|
> |m = 25|91.2|91.7|96.1|4:25:51|
> |m = 50|91.1|91.6|96.0|4:29:00|
> |m = 75|91.1|91.6|96.0|4:25:58|
>
>
> **Q4. How to ensure the selection of high-confidence samples from different classes?**
>
> To demonstrate that our method selects high-confidence samples across all classes (rather than biasing toward a few), we have shown how clustering precision and recall change across different classes before and after applying our method on CIFAR-10.
>
> **Precision(%):**
> |Method\Class|0|1|2|3|4|5|6|7|8|9|
> |---|---|---|---|---|---|---|---|---|---|---|
> |Propos|93.7|97.6|90.7|90.7|94.3|87.1|98.2|95.7|98.8|98.3|
> |Propos + Ours| **95.2**|**97.9**|**97.2**|**92.3**|**95.1**|**90.2**|**98.4**|**96.4**|98.7|**98.5**|
> **Recall(%):**
>
> |Method\Class|0|1|2|3|4|5|6|7|8|9|
> |---|---|---|---|---|---|---|---|---|---|---|
> |ProPos|97.5|95.6|93.2|87.0|97.3|88.3|98.7|97.0|92.8|97.4|
> |ProPos + Ours|**98.3**|**97.4**|**93.5**|**89.5**|**97.7**|**90.6**|**98.8**|**97.8**|**98.8**|**98.0**|
>
> Our method **improves the precision in 9 classes and recall in all 10 classes**. This confirms that our selection strategy does not favor specific classes, but instead **works consistently across diverse semantic groups.** The base model already yields relatively high clustering accuracy, and the local consistency sample selection is effective at identifying confident samples **within each semantic region** of the feature space. It **naturally achieves balanced coverage** across classes. The results also indicate that our method **helps refine the global feature space in a class-balanced manner.**
> We will include this analysis and the accompanying tables in the final version of the paper to further clarify this point.
>
>
> **Q5. Illustrating the trade-off through visualization.**
>
> Thank you for the constructive suggestion. We chose $k$ as the x-axis because our method adaptively selects the proper $k$ based on label consistency, and **the geometric interpretation provides an intuitive way to compare different $k$ values.** We agree that using precision as the x-axis could offer another interesting perspective to visualize the trade-off between quality and quantity, and we appreciate the suggestion and will add the figure in our final version.
>
>
> **Q6. Comparison with existing high-quality sample selection methods.**
>
> We have added new experiments comparing our method with several existing sample selection methods SSR[1],MOIT[2] and SCAN[3]. SSR and MOIT are **noise-robust sample filtering methods** and can directly replace our filtering mechanism, and SCAN is another sample selection **in deep clustering**. The evaluation covers clean sample accuracy, the number of true and false clean samples selected, and training time. Note that SCAN uses a confidence threshold to filter samples and is not directly applicable to the ProPos-like representation-based method. For a fair comparison, we treat SCAN as another baseline and evaluate both its original method and our sample selection method.
> As shown, our method is more conservative, aiming to minimize false positives. It achieves the **lowest number of incorrectly selected clean samples** while maintaining competitive accuracy.
>
> |Method|True Clean / Selected Clean|Accuracy(%)|False Clean Number|
> |---|--|---|---|
> |Propos + SSR|53978 / 55333|97.55|1355|
> |Propos + MOIT|29018 / 29750|97.54|732|
> |Propos + Ours |29720 / 30407|**97.74**|**687**|
> |SCAN |48609 / 49300|98.60|691|
> |SCAN + Ours|28458 / 28768|**98.92**|**310**|
>
> The computational overhead of sample selection alone is difficult to isolate; thus, we report **end-to-end training time**, including selection and model learning. We directly compare **the third stage of SCAN** and our method, and report **full SCAN runtime (stage 2 + 3)** in Appendix C, Table 11. Due to limited resources, SCAN experiments were run on an **RTX 3090**, while other methods were evaluated on **V100**.
>
> |Method|ACC(%)|Time (hour)|Device|
> |---|---|---|---|
> |ProPos|94.4|-|-|
> |ProPos+MOIT|94.4|-|-|
> |ProPos+SSR|92.5|-|-|
> |ProPos + Ours(MOIT selection)|95.8|5.0|V100|
> |ProPos + Ours(SSR selection)| 95.9|4.8|V100|
> |ProPos + Ours|**96.0**|**4.5**|V100|
> |SCAN|90.3|**3.7**|3090|
> |SCAN + Ours|**90.8**|3.8|3090|
>
> SSR and MOIT require full-dataset evaluation for sample selection, which increases the runtime. In contrast, our method leverages a **$k$-NN-based filter within each training batch**, enabling more efficient selection. Although SSR and MOIT improve performance when plugged into our training framework, such gains depend on our model design and loss functions. Used independently, these methods do not yield comparable performance, emphasizing the importance of integrating selection and training cohesively. Compared with SCAN, our method achieves a comparable efficiency but higher ACC.
>
> [1] Multi-objective interpolation training for robustness to label noise. CVPR, 2021.
>
> [2] Ssr: An efficient and robust framework for learning with unknown label noise. BMVC, 2021.
>
> [3] Scan: Learning to classify images without labels, ECCV, 2020.

---

> > ### Author Response · Authors · 2025-08-04
> >
> > Dear Reviewer **EzRr**,
> >
> > Thanks again for your time and effort in reviewing this paper. We appreciate your valuable comments to improve the quality of this work. If you still have concerns about this work, we would be more than happy to clarify them. Thanks.
> >
> > Regards from the authors.

---

> > > ### Author Response · Authors · 2025-08-06
> > >
> > > Dear Reviewer EzRr,
> > >
> > > Thanks again for your time and effort in reviewing this paper. We appreciate your valuable comments to improve the quality of this work.  **As the author-reviewer discussion period deadline is approaching**, please take five to ten minutes to check whether your concerns have been addressed.
> > >
> > > Thanks.

---

> > ### Comment · Reviewer_EzRr · 2025-08-07
> >
> > Thank you for your response—my primary concerns have been resolved, and I will update my rating accordingly.

---

> > > ### Author Response · Authors · 2025-08-07
> > >
> > > Dear **EzRr**,
> > >
> > > We are glad that your primary **concerns have been resolved**. Thanks for **raising your rating**. We will follow your suggestions to revise this paper in the final version. Thanks.
> > >
> > > Regards.

---

### Note · Authors · 2025-08-13

We thank reviewers for the constructive feedback and recognition of our contributions. We sincerely appreciate that you have found our work:


- proposes a novel (Reviewer EzRr) and valuable (Reviewer S4wz) **adaptive k-NN mechanism** for high-confidence sample selection (Reviewer Tv3M), effectively balancing the quality and quantity of self-supervision (Reviewer EzRr).

- offers a **parameter-free, plug-and-play** design without modifying existing architectures (Reviewers EzRr, S4wz, ZuXL, Tv3M), which is simple, efficient, and easily integrated into real-world and legacy systems (Reviewer S4wz).

- leverages **trustworthy local cues** to optimize **global cluster structures** (Reviewers EzRr, S4wz, ZuXL), described as intuitive, novel, and practically relevant (Reviewer S4wz), enhancing **intra-class compactness** and **inter-class separability** via selective instance- and cluster-level losses (Reviewer Tv3M).

- is effective and generalizable (Reviewers EzRr, ZuXL), achieving **consistent** (Reviewer Tv3M), **effective** (Reviewer ZuXL), and **significant** (Reviewer EzRr) improvements (Reviewer S4wz) over state-of-the-art baselines across diverse datasets.

- includes **comprehensive ablation studies** validating each design choice (Reviewer S4wz).


Regarding concerns on **method design** and **experimental setup**, we have provided further clarifications that effectively address these points.

For issues related to **data imbalance**, **challenges in datasets with many classes**, and **comparisons with other methods**, we conducted additional experiments that were well-received by reviewers, resulting in positive feedback. We will incorporate these new results and corresponding analyses into the final version of the paper.

In summary, our method introduces a **parameter-free plug-in** that exploits **high-quality local structural cues** to identify **high-confidence samples**, directly guiding **global structure optimization**. This well-motivated integration of local reliability into global clustering objectives is **novel**, requires **no architectural modifications**, and is highly practical for deployment across diverse existing models. Extensive experiments demonstrate that it **consistently delivers substantial** performance improvements over a wide range of deep clustering methods.

Regards from the authors.

---

### Decision · Program_Chairs · 2025-09-17

**Decision:**

Accept (poster)

**Comment:**

This paper presents DCBoost, a novel deep clustering enhancement method for improving clustering performance by refining the global structure of existing deep clustering models. One of the clear (practical) advantages of this approach is that it can be plugged into existing clustering models and doesn’t require hyper-param tuning. The authors also demonstrate the gains from this technique when combined with existing clustering models evaluated across multiple datasets.

Overall, the paper is interesting and the authors provide good clarity & additional details in the discussion that address many of the reviewer concerns. There are several strengths already highlighted by the reviewers. A few of the weaknesses (also ack’d by authors) is the performance degradation under different settings (eg high imbalance, non-uniform data dist.) The authors provide some (limited) experiment results in the discussion to address this. If anything, one of the areas that could benefit from additional work is extending the empirical evaluations to large(r) benchmark datasets and going beyond simple, image datasets (e.g, language tasks with global structure) — there are several publicly available at this point. It would help further strengthen the empirical claims as well as practical applicability of the proposed approach. The authors are also encouraged to incorporate all feedback from the reviews+discussions in the final version of the paper.